# Repulsive Monte Carlo On The Sphere For The Sliced Wasserstein Distance

**Vladimir Petrovic**                                                    *vladimir.petrovic@univ-lille.fr*
*Univ. Lille, CNRS, Centrale Lille, UMR 9189 - CRIStAL, 59651 Villeneuve d'Ascq, France*

**Rémi Bardenet**                                                        *remi.bardenet@univ-lille.fr*
*Univ. Lille, CNRS, Centrale Lille, UMR 9189 - CRIStAL, 59651 Villeneuve d'Ascq, France*

**Agnès Desolneux**                                                  *agnes.desolneux@ens-paris-saclay.fr*
*Univ. Paris-Saclay, ENS Paris-Saclay, CNRS, Centre Borelli UMR 9010, 91190 Gif-sur-Yvette, France*

**Reviewed on OpenReview:** *https: // openreview. net/ forum? id= JSiTmB6Ehu*

## Abstract

In this paper, we consider the problem of computing the integral of a function on the unit sphere, in any dimension, using Monte Carlo methods. Although the methods we present are general, our guiding thread is the sliced Wasserstein distance between two measures on $\mathbb{R}^d$, which is precisely an integral of the $d$-dimensional sphere. The sliced Wasserstein distance (SW) has gained momentum in machine learning either as a proxy to the less computationally tractable Wasserstein distance, or as a distance in its own right, due in particular to its built-in alleviation of the curse of dimensionality. There has been recent numerical benchmarks of quadratures for the sliced Wasserstein (Sisouk et al., 2025), and our viewpoint differs in that we concentrate on quadratures where the nodes are repulsive, i.e. negatively dependent. Indeed, negative dependence can bring variance reduction when the quadrature is adapted to the integration task. Our first contribution is to extract and motivate quadratures from the recent literature on determinantal point processes (DPPs) and repelled point processes, as well as repulsive quadratures from the literature specific to the sliced Wasserstein distance. We then numerically benchmark these quadratures. Moreover, we analyze the variance of the *UnifOrtho* estimator, an orthogonal Monte Carlo estimator introduced by Rowland et al. (2019). Our analysis sheds light on *UnifOrtho*'s success for the estimation of the sliced Wasserstein in large dimensions, as well as counterexamples from the literature. Our final recommendation for the computation of the sliced Wasserstein distance is to use randomized quasi-Monte Carlo in low dimensions and *UnifOrtho* in large dimensions. DPP-based quadratures only shine when quasi-Monte Carlo also does, while repelled quadratures show moderate variance reduction in general, but more theoretical effort is needed to make them robust.

## 1 Introduction

In Monte Carlo integration, introducing repulsion between the points at which the integrand is evaluated can bring a significant variance reduction. In $\mathbb{R}^d$, determinantal point processes (DPP) have for example been shown to yield a central limit theorem with improved convergence rate over classical Monte Carlo, for compactly supported integrands (Bardenet & Hardy, 2020; Coeurjolly et al., 2021). In the same vein, even a modicum of negative dependence can reduce variance, e.g. applying a single step of a gradient descent aimed at minimizing the Coulomb energy between the quadrature nodes (Hawat et al., 2023). Beyond Euclidean spaces, Monte Carlo methods with DPPs have been considered over selected manifolds (Berman, 2024; Lemoine & Bardenet, 2024). One natural manifold to look at is the sphere $\mathbb{S}^{d-1} \subset \mathbb{R}^d$; however,

beyond the case of $\mathbb{S}^2$ treated in Berman (2024), it is not yet clear whether DPPs and similar randomized quadratures with negative dependence can be a practical asset.

In machine learning, the problem of integrating over $\mathbb{S}^{d-1}$ naturally arises in recent applications of optimal transport. A central object in optimal transport is the so-called Wasserstein distance, an intuitive distance between probability measures with a host of theoretical properties (Peyré & Cuturi, 2018). On the negative side, numerically evaluating the Wasserstein distance between two measures typically starts with replacing these two measures by i.i.d. realizations, but the quality of the approximation rapidly degrades with the dimension (Fournier & Guillin, 2015). Moreover, even between two discrete distributions with $M$ atoms each, the cost of a generic algorithm to compute the Wasserstein distance scales as $M^3 \log(M)$, which becomes intractable for large $M$ (Peyré & Cuturi, 2018). This has led to research on alternatives to the Wasserstein distance, one of which is the sliced Wasserstein distance (SW).

The sliced Wasserstein distance finds its roots in one-dimensional optimal transport (Bonnotte, 2013). The cost of computing the Wasserstein distance between two discrete distributions with their atoms on a line essentially boils down to sorting the abscissa of the atoms. In higher dimension $d$, the idea is hence to look at the projection of our discrete measures on a given direction, compute the Wasserstein distance between these projected point clouds, and integrate the results along all possible directions. The corresponding quantity is an integral over the sphere, the integrand being a one-dimensional Wasserstein distribution, that defines a metric over the space of probability measures called the sliced Wasserstein distance. The SW distance preserves the main topological properties of the Wasserstein distance, while holding the promise to solve the aforementioned curse of dimensionality and tractability issues (Bayraktar & Guo, 2021; Nadjahi, 2021).

The SW distance has found many applications in machine learning, in gradient descent (Bonet et al., 2022), barycenter computation (Bonneel et al., 2015), generative models (Deshpande et al., 2018; Liutkus et al., 2019) or kernel methods (Kolouri et al., 2016). It also finds applications, for instance, in texture synthesis (Heitz et al., 2021), in regularizing autoencoders (Kolouri et al., 2019) or differential privacy (Rakotoma-monjy & Liva, 2021). More recently, it has also been applied to gradient descent on manifolds (Bonet et al., 2025). Finally, the SW distance has also been used as a proxy to the Wasserstein distance, when comparing the output of different sampling algorithms (Linhart et al., 2024). All the aforementioned applications use repeated evaluations of the sliced Wasserstein distance, e.g. between a reference measure and an iteratively refined approximation. Errors in the evaluations of the SW distance can thus deteriorate the performance of the underlying ML tasks, and it is natural to try to understand and reduce these evaluation errors.

When computing the sliced Wasserstein distance, two kinds of approximations are actually being made. The first one derives from the fact that one usually has access only to a finite sample from the underlying probability distributions; this is known as the sample complexity problem (Manole et al., 2022). The second kind of error comes from the formulation itself of the sliced Wasserstein distance between two finitely supported measures as an integral over the sphere, which has to be approximated in general. The present paper focuses on that second kind of error, which is intuitively independent of the initial ML task. One typically relies on Monte Carlo algorithms on the sphere to get an estimate of the desired quantity. Although the cost of evaluating the integrand is relatively cheap, stacking up a large number $N$ of evaluations on as many directions on the sphere can still become computationally heavy, and the slow decay in $N^{-1/2}$ of the error of crude Monte Carlo integration will typically require such a large $N$ (Robert & Casella, 2004).

Several Monte Carlo methods have already been investigated to solve the integration task inherent to computing the SW distance. In particular, while we were working on this manuscript, a survey has appeared (Sisouk et al., 2025). Their conclusions are that for $d \in \{2, 3\}$, quasi-Monte Carlo methods prevail, while in higher dimension (typically above $d = 20$), the so-called *orthogonal Monte Carlo* method (Rowland et al., 2019; Lin et al., 2020) is both more efficient than crude Monte Carlo and computationally cheap enough to be practical in ML applications. In the intermediate range, they do not provide clear guidelines but rather encourage the reader to experiment. Besides also reviewing existing Monte Carlo methods for SW estimation, our contributions are twofold. First, we introduce and benchmark five randomized quadratures that have not yet been used to estimate the sliced Wasserstein distance. One of these is a natural importance sampling baseline. The four others are joint distributions with negative dependence that we draw and sometimes

mildly adapt from the recent literature on repulsive Monte Carlo methods. Some of the resulting estimators already provably enjoy faster decaying variance than i.i.d. quadratures. To our knowledge, when considering the sliced Wasserstein distance, this has only been achieved by the estimator from Leluc et al. (2024). On top of the interest of computing the sliced Wasserstein distance, our numerical investigations are also meant to help us identify repulsive point processes that are useful for Monte Carlo integration on the sphere. Indeed, proving a variance reduction result with negative dependence as in (Bardenet & Hardy, 2020; Hawat et al., 2023) can be long and technical, so that it is important that the community focus their mathematical efforts on the most promising candidates. Precisely doing that, i.e. focussing our mathematical efforts on understanding practically successful estimators, our second main contribution is to compute the variance of an estimator based on orthogonal Monte Carlo (Rowland et al., 2019; Lin et al., 2020). The latter has already been empirically shown to be successful for SW estimation in large dimensions, which our own experiments confirm. Our variance calculation sheds light on the situations where orthogonal Monte Carlo may (or may not) yield variance reduction.

The rest of the paper is organized as follows. Section 2 introduces background on repulsive point processes for Monte Carlo integration. Section 3 describes the main properties of the sliced Wasserstein distance, and reviews numerical quadratures that have already been implemented to estimate it. Section 4 presents new candidate estimators for the sliced Wasserstein distance, among which a natural importance sampling scheme and various repulsive point processes adapted to the spherical case. Section 5 presents our derivation of the variance of an orthogonal Monte Carlo estimator known as *UnifOrtho*. All these methods are empirically evaluated and benchmarked in Section 6. Section 7 concludes the paper. The appendix presents supplementary background on spherical harmonics, additional details on the importance sampling baseline, as well as additional experiments.

## 2 Repulsive Monte Carlo

Monte Carlo methods are randomized algorithms for quadrature, i.e., numerical integration. The common idea is to build linear combinations of a finite number of integrand evaluations at well-chosen quadrature *nodes* (Robert & Casella, 2004). While classical Monte Carlo methods draw their nodes using independent random variables or a Markov chain, many recent works have tried to leverage negative dependence among nodes in $\mathbb{R}^d$ to obtain lower mean-square integration errors. One family of approaches focuses on settings where the target is hard to sample from or approximate, and introduces negative dependence into Monte Carlo Markov chains. For instance, the Wang-Landau algorithm is an adaptive MCMC algorithm that forces the chain to move away from overexplored regions (Fort et al., 2015). A more recent example is (Ye et al., 2020), where the authors introduce in a Langevin diffusion a term that pushes the state away from the past history of the chain, without altering the limiting distribution. However, so far, these algorithms only come with asymptotic guarantees, and the corresponding rates of convergence are still the same as without the negative dependence. A second family of approaches considers the more restricted setting where the target is easy to sample from or approximate and the dimension is low, and manages to improve on the convergence rate thanks to repulsiveness. In this setting, the bottleneck comes from the number of calls to the integrand, and is already at the center of a vast and rich literature, see e.g. (Delyon & Portier, 2016; Leluc et al., 2025). We focus in this paper on this restricted setting where the target is easy to sample from, and we review two recent ways to introduce negative dependence in integration routines in $\mathbb{R}^d$, determinantal and repelled point processes. We choose these two because they easily adapt to the sphere, as we shall see in Section 4.

### 2.1 Determinantal point processes

Initially invented to model the arrival times of physical fermions in optics (Macchi, 1972), determinantal point processes (DPPs) have seen a recent surge of interest in probability (Soshnikov, 2000; Hough et al., 2006), statistics (Lavancier et al., 2015), and machine learning (Kulesza & Taskar, 2012). Formally, we shall only use *projection DPPs*, which can be defined as follows.

**Definition 1 (Projection DPP)** *Let $\mathbb{X}$ be a separable complete metric space, and $\mu$ be a measure on its Borel sets. Let $N \geq 1$, and $\phi_0, \ldots, \phi_{N-1}$ be orthonormal functions in $L^2(\mu)$. Let*

$$K : x, y \mapsto \sum_{k=0}^{N-1} \phi_k(x)\phi_k(y). \tag{1}$$

*Let $(X_1, \ldots, X_N)$ be drawn from*

$$\frac{1}{N!} \det((K(x_i, x_j))_{1 \leq i,j \leq N}) \, d\mu(x_1) \ldots d\mu(x_N). \tag{2}$$

*Then, we say that the random set $X = \{X_1, \ldots, X_N\} \subset \mathbb{X}$ has for distribution the projection DPP of kernel $K : \mathbb{X} \times \mathbb{X} \to \mathbb{R}$ and reference measure $\mu$, and we write $X \sim \mathrm{DPP}(K, \mu)$.*

First, we note that (2) defines a *bona fide* probability distribution because $K$ in (1) is a projection kernel, namely the kernel of the projection onto $\mathrm{Span}(\phi_0, \ldots, \phi_{N-1})$; see e.g. Hough et al. (2006). Second, DPPs are repulsive in the sense that the determinant in (2) favors configurations where the $X_i$s spread evenly across $\mathbb{X}$. Indeed, if $K$ is smooth, two points close to each other correspond to two nearly identical columns in the Gram matrix $((K(x_i, x_j))_{1 \leq i,j \leq N})$, and thus a small determinant. Third, DPPs with non-projection kernels can be defined (Hough et al., 2006), but we shall only be concerned by projection kernels in this paper. Finally, on top of having a relatively simple expression, a computational advantage of DPPs that makes them an ideal candidate for summarization tasks is that the chain rule for (2) can be simply expressed using Schur complements (Hough et al., 2006)[Proposition 19]. In more detail, to sample (2), it is enough to sample $X_1$ from $1/N \cdot K(x_1, x_1)d\mu(x_1)$, and for $k = 2, \ldots, N$, iteratively sample $X_k$ from

$$\frac{K(x_k, x_k) - K(x_k, x_{1:k-1})\mathbf{K}_{k-1}^{-1}K_{k-1}(x_{1:k-1}, x_k)}{N - k + 1}d\mu(x_k), \tag{3}$$

where $K(x_k, x_{1:k-1}) = K(x_k, x_{1:k-1})^T$ is short for $(K(x_k, x_1), \ldots, K(x_k, x_{k-1}))$, and $\mathbf{K}_{k-1} = (K(x_i, x_j))_{1 \leq i,j \leq k-1}$. Individual sampling steps in (3) are typically implemented using rejection sampling. In Gautier et al. (2019b), the total number of rejections for sampling an Orthogonal Polynomial Ensemble in $\mathbb{R}^d$ is estimated to be $\mathcal{O}(2^d N \log(N))$. A realization of this particular DPP in the square $[-1, 1]^2$ can be observed in Figure 1c. In general, much is known on sampling DPPs, exactly or approximately (Gautier, 2020; Barthelmé et al., 2023).

## 2.2 Monte Carlo integration with DPPs

Monte Carlo methods relying on DPPs with specific kernels have been investigated when $\mathbb{X} = \mathbb{R}^d$ and the target measure has a density w.r.t. the Lebesgue measure, e.g. (Bardenet & Hardy, 2020; Mazoyer et al., 2020; Belhadji et al., 2019; 2020). A general conclusion is that for the right choice of kernel, a DPP with cardinality $N$ can integrate smooth functions with a mean squared error in $o(1/N)$, thus decaying faster than for classical Monte Carlo methods. For instance, the so-called multivariate orthogonal polynomial ensembles studied in (Bardenet & Hardy, 2020) yield a mean squared error in $1/N^{1+1/d}$ for integrands that are continuously differentiable. In this paper, we rather consider integration on the sphere $\mathbb{S}^{d-1} \subset \mathbb{R}^d$. Using a change of variables such as spherical coordinates, it is straightforward to adapt e.g. the quadratures proposed by (Bardenet & Hardy, 2020; Mazoyer et al., 2020) for $[-1, 1]^{d-1}$ to $\mathbb{S}^{d-1}$, at the price of an artificial accumulation of points. Closer to our interest for the sphere, Lemoine & Bardenet (2024) show that for a compact complex manifold of complex dimension $d/2$ (and thus dimension $d$ when seen as a real manifold), the right choice of kernel in (2) yields the faster rate $1/N^{1+2/d}$. This applies to $\mathbb{S}^2$, where the corresponding DPP is called the *spherical ensemble*; see Section 4.2.2 for more details. However, this result does not easily generalize to $\mathbb{S}^{d-1}$ with $d > 3$. Still in the particular case $d = 3$, even finer results are available in (Berman, 2024), actually the first paper to explicitly investigate a DPP for integration on the sphere. Berman (2024) provides a theoretical analysis of the worst-case integration error of the spherical ensemble for functions on the sphere in specific Sobolev classes. Finally, for integrands that are smooth enough to belong to a reproducing kernel Hilbert space (RKHS), DPPs (Belhadji et al., 2019) and mixtures of DPPs (Belhadji et al., 2020; Belhadji, 2021) have been proven to yield fast-decaying mean squared errors.

### 2.3 Quadratic-time alternatives to DPPs

Sampling a DPP, while polynomial, can still be long enough to discard DPPs when the cost of evaluating the integrand is low. In particular, one needs to come up with rejection sampling routines to sample the conditionals (3) in a reasonable time; see (Gautier et al., 2019a) for a discussion. Aside from DPP sampling, there are $\mathcal{O}(N^2)$ algorithms that can still achieve a mean squared error decay in $o(1/N)$. For instance, Delyon & Portier (2016) propose a variant of importance sampling where the proposal PDF is replaced by a kernel density estimator, with a fast error decay. A particularly natural repulsive strategy that does not require strong smoothness assumptions on the integrand is known as *repelled point processes* (Hawat et al., 2023). The idea is to draw a computationally cheap randomized quadrature, and apply one step of a gradient descent aimed at minimizing the Coulomb energy of the configuration of quadrature nodes, as if they were identically charged particles. The result of such a procedure can be observed in Figure 2. It is easy to come up with a similar algorithm for points on the sphere, as we shall do in Section 4. However, the main variance reduction result of Hawat et al. (2023) does not hold for the sphere, and we see our paper as an exploration of which algorithms have promising empirical performances to motivate their theoretical study.

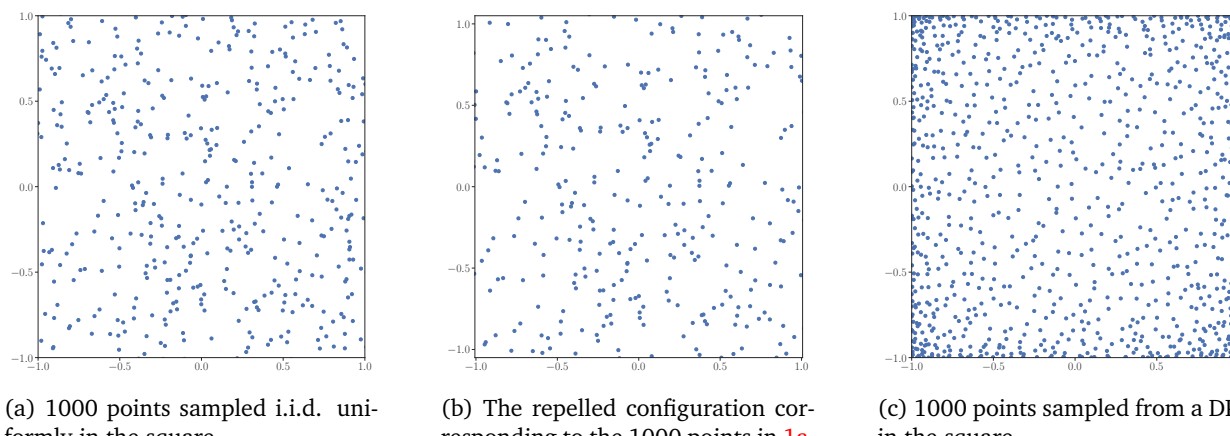

(a) 1000 points sampled i.i.d. uniformly in the square.

(b) The repelled configuration corresponding to the 1000 points in 1a.

(c) 1000 points sampled from a DPP in the square.

Figure 1: Realizations of three point processes

## 3 The sliced Wasserstein distance

Our motivating application for integration on the sphere is the computation of the sliced Wasserstein (SW) distance between two probability distributions.

**Definition 2** *Let $d \in \mathbb{N}$, $p > 0$, and $\mu, \nu$ be two probability measures on $\mathbb{R}^d$. The sliced Wasserstein distance between $\mu$ and $\nu$ is*

$$SW_p(\mu, \nu) = \left( \int_{\mathbb{S}^{d-1}} [W_p(\theta_\# \mu, \theta_\# \nu)]^p \, \mathrm{d}\theta \right)^{1/p}, \tag{4}$$

*where $\theta_\# \mu$ denotes the push-forward measure of $\mu$ by the function $a_\theta : x \in \mathbb{R}^d \to \theta^T x \in \mathbb{R}$, $W_p$ is the one-dimensional $p$-Wasserstein distance, and the integral is with respect to the uniform measure on the sphere.*

Before discussing its evaluation cost, we give some motivating facts about the SW distance, and refer to Nadjahi (2021) for an exhaustive reference.

### 3.1 Motivating properties

First, for all $p \geq 1$, $SW_p$ metrizes the weak convergence on the space of finite $p$-moments probability measures (Nadjahi, 2021) [Theorem 3.1]. On compact domains, the topology induced by the sliced Wasserstein

metric is actually equivalent to the one induced by the Wasserstein metric since, if $\mu$, $\nu$ are supported on $B(0, R)$,

$$SW_p(\mu, \nu) \leq W_p(\mu, \nu) \text{ and } W_p^p(\mu, \nu) \leq C_{d,p} R^{(p-1)/(d+1)} SW_p(\mu, \nu)^{1/(d+1)}, \tag{5}$$

where $C_{d,p}$ is a constant only depending on $p$ and $d$ (Bonnotte, 2013)[Proposition 5.1.3 and Theorem 5.1.15]. Second, and importantly to train e.g. generative models, it is also possible to perform gradient descent on the SW metric (Nguyen et al., 2024). Formally, if for $\mathbf{X} \in \mathbb{R}^{d \times M}$, $\mu_{\mathbf{X}}$ denotes the empirical measure supported on the columns of $\mathbf{X}$, then for a discrete measure $\nu$, the map $\mathbf{X} \in \mathbb{R}^{d \times M} \to SW_2^2(\mu_{\mathbf{X}}, \nu) \in \mathbb{R}$ is $\mathcal{C}^1$ (Bonneel et al., 2015) [Theorem 1].

Focussing now on its practical evaluation, one key advantage of the SW distance over the Wasserstein distance is its dimension-free sample complexity. Formally, for a measure $\mu$, write its $p$-moment as $m_p(\mu) = \int \|t\|^p d\mu(t)$ and $\hat{\mu}_M$ for the empirical measure obtained from $M$ i.i.d. draws from $\mu$. For $p \in [1, \infty)$, assume that $\mu$ and $\nu$ both have a finite moment of order $q > p$. Then

$$\mathbb{E}[|SW_p(\hat{\mu}_M, \hat{\nu}_M) - SW_p(\mu, \nu)|] \leq C_{pq}^{1/q} m_q^{1/q}(\mu, \nu) \times \begin{cases} M^{-1/2p} \text{ if } q > 2p \\ M^{-1/2p} \log(M)^{1/p} \text{ if } q = 2p \\ M^{-(q-p)/pq} \text{ if } q \in (p, 2p) \end{cases},$$

where $m_q^{1/q}(\mu, \nu) = m_q^{1/q}(\mu) + m_q^{1/q}(\nu)$ (Nadjahi, 2021) [Theorem 7.14]. In particular, it is enough to focus on evaluating the SW distance (4) between two empirical measures of support of cardinality $M$. This is an integral over the sphere, which cannot be solved analytically, thus requiring numerical quadrature. Fortunately, the integrand can be efficiently computed: for $p \geq 1$, it can be done exactly in time $\mathcal{O}(M \log M + Md)$. The $Md$ part comes from the computation of the projection, as each point on which the measure is supported has to be projected onto a line. As for the $M \log(M)$ part, it comes from the theory of one-dimensional optimal transport, where one can show that the only computational bottleneck is essentially sorting the atoms of the involved (discrete) measures (Peyré & Cuturi, 2018, Remark 2.30).

The choice of a numerical quadrature can be informed by the smoothness of the integrand. We know that the map

$$f_{\mu,\nu}^{(p)} : \theta \in \mathbb{S}^{d-1} \to W_p^p(\theta_\# \mu, \theta_\# \nu), \tag{6}$$

is Lipschitz (Bayraktar & Guo, 2021)[Proposition 2.2], with Lipschitz constant

$$p W_p(\mu, \nu)^{p-1}(m_p(\mu)^{1/p} + m_p(\nu)^{1/p}).$$

However, when both measures are discrete, while the map (6) is $\mathcal{C}^\infty$ outside of a set of measure $0$ on the sphere, it fails to be globally $\mathcal{C}^1$ in general. This motivates the use of numerical quadratures that do not make strong smoothness assumptions on the integrand, intuitively dismissing methods that rely on the target integrand being in an RKHS such as (Belhadji et al., 2019; 2020). Moreover, the dependence in $Md$ of the cost of evaluating the integrand –due to computing the projections in the push-forward measures– justifies searching for quadratures with a fast-decaying error if we are to estimate the SW between large datasets in high-dimensional spaces. Repulsive Monte Carlo methods such as DPP-based quadratures and repelled point processes thus seem natural to investigate. Before doing so, we quickly review the existing literature on advanced Monte Carlo techniques for the SW.

### 3.2 Existing Monte Carlo methods for the sliced Wasserstein distance

Besides the natural i.i.d. sampling on the sphere, several advanced Monte Carlo methods have been proposed that reduce the mean squared error in estimating (4), using either control variates or randomized grids.

#### 3.2.1 Control variates

Control variates is a standard variance reduction technique in Monte Carlo integration (Owen, 2013, Chapter X). In a nutshell, consider $\varphi_i : \mathbb{S}^{d-1} \to \mathbb{R}$, $i = 1, \ldots, s$, such that $\int \varphi_i(\theta) d\theta = 0$ for all $i$. Letting $f : \mathbb{S}^{d-1} \to \mathbb{R}$ be a square-integrable function, and $\theta_1, \ldots, \theta_N$ be drawn i.i.d. uniformly on the sphere, consider the ordinary least-squares (OLS) problem

$$\left(I_N^{\mathrm{ols}}(f), \beta_N^{\mathrm{ols}}(f)\right) = \operatorname*{argmin}_{\alpha \in \mathbb{R}, \beta \in \mathbb{R}^s} \left\{ \sum_{i=1}^N (f(\theta_i) - \alpha - \sum_{j=1}^s \beta_j \varphi_j(\theta_i))^2 \right\}. \tag{7}$$

To gain intuition, we note that for a fixed $\beta \in \mathbb{R}^s$, minimizing the RHS of (7) in $\alpha$ yields the empirical mean of $f(\theta_i) - \sum_{j=1}^s \beta_j \varphi_j(\theta_i)$, where $i = 1, \ldots, N$. This should in turn be close to $\int f(\theta) \mathrm{d}\theta$ since the $\varphi_j$ have integral zero. Optimizing over $\beta$ further reduces the variance of $I_N^{\mathrm{ols}}(f)$ at the cost of introducing a small bias. The key decision to be made by the practitioner is the choice of $s$ and the *control variates* $\varphi_1, \ldots, \varphi_s$. In particular, as both $s$ and $N$ go to infinity, if the space spanned by the control variates is large enough to allow reconstructing the integrand $f$, Portier & Segers (2019) obtain a central limit theorem with a squared error decaying faster than the Monte Carlo rate $1/N$. We now present two choices of control variates that have been proposed in the specific case of the sliced Wasserstein integrand (6): the *up/low* method of Nguyen & Ho (2024) and the spherical harmonics in (Leluc et al., 2024).

**Control variates "up" and "low".** For two probabilities $\mu, \nu$ on $\mathbb{R}^d$ with finite first and second moments $m_\mu, m_\nu, \Sigma_\mu, \Sigma_\nu$, we know (Peyré & Cuturi, 2018, Remark 2.9) that the 2-Wasserstein distance satisfies

$$W_2^2(\mu, \nu) = \|m_\mu - m_\nu\|^2 + W_2^2(\tilde\mu, \tilde\nu), \tag{8}$$

where $\tilde\mu$, $\tilde\nu$ are the centered versions of $\mu$ and $\nu$, ie $\tilde\mu = t_\#^{(\mu)}\mu$, where $t^{(\mu)} : x \in \mathbb{R}^d \to x - m_\mu$. When computing $SW_2$, Nguyen & Ho (2024) thus suggest taking $s = 1$ control variate in (7), with $\varphi_1$ equal to

$$\varphi_{\mathrm{low}} : \theta \mapsto \left(\theta^T(m_\mu - m_\nu)\right)^2 - \frac{1}{d}\|m_\mu - m_\nu\|^2. \tag{9}$$

Note that $\varphi_{\mathrm{low}}$ is centered, and that it will likely have little impact when either $p \neq 2$ or the target distributions are already centered. In the same spirit, Nguyen & Ho (2024) also propose $s = 1$, with $\varphi_1$ this time equal to

$$\varphi_{\mathrm{up}} : \theta \mapsto \varphi_{\mathrm{low}}(\theta) + \theta^T \Sigma_\mu \theta + \theta^T \Sigma_\nu \theta - \frac{1}{d}\left(\mathrm{Tr}(\Sigma_\mu) + \mathrm{Tr}(\Sigma_\mu)\right), \tag{10}$$

where the quadratic term on top of $\varphi_{\mathrm{low}}$ upper-bounds the $W_2$ distance between two centered Gaussians –hence the label *up*– and the remaining term guarantees a null integral, as required for a control variate. This time, the control variate is expected to pick up second-order information. Finally, note that while $\varphi_{\mathrm{low}}$ and $\varphi_{\mathrm{up}}$ use rather crude approximations to the integrand and are limited to the case $p = 2$, they are both cheap to compute and provide useful baselines.

**Spherical Harmonics.** Still based on (7), Leluc et al. (2024) rather propose to take $1, \varphi_1, \varphi_2, \ldots$ to be spherical harmonics $\{Y_k^\ell, \ell \geq 0, 1 \leq k \leq h_\ell\}$, ordered in the lexicographic order of $(\ell, k)$. To wit, $Y_0^0 = 1$ is constant, and, for $\ell \geq 1$, $\{Y_k^\ell, \ell \geq 1\}$ form an orthonormal basis of $\mathcal{H}_\ell$, the $h_\ell$-dimensional set of harmonic homogeneous polynomials of degree $\ell$ restricted to $\mathbb{S}^{d-1}$. We refer to (Leluc et al., 2024, Section 4.1) or our Appendix A.1 for a quick self-contained definition of spherical harmonics, but for now it suffices to say that $1, \varphi_1, \varphi_2, \ldots$ is an orthonormal basis of $L^2(\mathbb{S}^{d-1})$. For a fixed $N$, let $s = s_N$ be the number of spherical harmonics of degree at most $2L_N$. Note that $s_N = \mathcal{O}(L_N^{d-1})$. The estimator $SHCV_N^p(\mu, \nu)$ of the SW distance between two probability measures on $\mathbb{R}^d$ is then defined to be $I_N^{\mathrm{ols}}$ in (7).

Leluc et al. (2024) prove that for $d \geq 2$, $p \geq 1$, $\mu, \nu$ having finite $p$-th moments, and when $s_N = o(N^2)$, so that $L_N = N^{1/2(d-1)}/\ell_N$ for some sequence $\ell_N$ going arbitrarily slowly to $+\infty$ when $N$ grows,

$$|SHCV_N^p(\mu, \nu) - SW_p^p(\mu, \nu)| = \mathcal{O}_{\mathbb{P}}(\ell_N N^{-(1/2 + 1/2(d-1))}), \tag{11}$$

demonstrating a reduction in the error rate compared to standard Monte Carlo. The whole procedure runs in $\mathcal{O}(N\omega_f + Ns_N^2 + s_N^3)$, where $\omega_f$ is the time complexity of evaluating $f$, so that the procedure is quadratic if $s_N = o(N^2)$ as prescribed. It is expected that $SHCV_N^p$ will be efficient when the integrand (6) appearing in the definition of the SW distance will be well-approximated by polynomials of degree lower than $s_N$,

and that it will outperform the control variates $\varphi_{\text{low}}$ and $\varphi_{\text{up}}$ as soon as the degree is large enough, at a higher computational price, however. Another caveat that we shall discuss again later is that the complexity estimate ignores the computational time spent evaluating spherical harmonics, which can be prohibitive in large-dimensional settings; see also Appendix A.1.

### 3.2.2 Randomized grids

Letting $N$ be the number of evaluations of the integrand (6) that one is willing to spend, and assuming for simplicity that $k = N/d$ is an integer, Rowland et al. (2019) propose to take $k$ i.i.d draws from the Haar measure on the orthogonal group $O(d)$. The columns of these matrices are then marginally uniformly distributed on the sphere, and the average of the integrand (6) over the reunion of these $N = kd$ columns is thus an unbiased estimator, called the *UnifOrtho* estimator in (Rowland et al., 2019). Intuitively, since the columns of a single Haar draw are orthonormal, they fill the sphere quite evenly, thus justifying our classification as a randomized grid. One could expect some variance reduction coming from this very uniform spread, but there appears to be no such theoretical guarantee so far. Rowland et al. (2019) even exhibit a counterexample of two empirical measures such that *UnifOrtho* yields a worse (i.e. higher-variance) SW estimator than crude i.i.d. Monte Carlo on the sphere. We clarify the situation with an explicit derivation of the variance of the *UnifOrtho* estimator in Section 5.

Quasi-Monte Carlo (QMC; Dick & Pilichshammer, 2010) methods are deterministic quadratures that can be thought of as the computationally tractable higher-dimensional version of a grid. Worst-case guarantees usually involve proving that the quadrature nodes have *low discrepancy*, and an additional randomization can help obtain guarantees with more tractable constants. QMC methods for computing the SW distance have been numerically investigated in the three-dimensional setting in Nguyen et al. (2024). However, there is no known low-discrepancy sequence on $\mathbb{S}^{d-1}$, as soon as $d \geq 3$. An empirically promising alternative (Nguyen et al., 2024; Sisouk et al., 2025) is to use the so-called *Fekete points* as quadrature, a notion from potential theory defined as the set of points that minimize a particular interaction potential over the sphere. We note however that the construction of Fekete points on the sphere in polynomial time is known to be a hard problem, and in dimension 3 is even listed as Smale's 7th problem (Smale, 1998). A more straightforward alternative to low-discrepancy quadratures is to map a low-discrepancy sequence in $[0,1]^{d-1}$ to $\mathbb{S}^{d-1}$, via some transformation such as using the inverse cumulative function of the normal distribution. Empirical results have been however less encouraging (Nguyen et al., 2024; Sisouk et al., 2025).

In this paper, we will consider a randomized QMC benchmark in two and three dimensions, i.e. on $\mathbb{S}^1$ and $\mathbb{S}^2$. On $\mathbb{S}^2$, we use the *generalized spiral points* from Rakhmanov et al. (1994), which are easy to draw and have been proven to have low discrepancy, at least asymptotically (Brauchart et al., 2014). To wit, consider $z_i = 1 - (2i - 1)/N$ for $1 \leq i \leq N$ and

$$\Phi_{i,1} = \cos^{-1}(z_i), \; \Phi_{i,2} = 1.8\sqrt{N}\Phi_{i,1} \bmod(2\pi). \tag{12}$$

The generalized spiral points are the points on the sphere with spherical coordinates $(\Phi_{i,1}, \Phi_{i,2})$. Note that the constant $1.8$ is chosen arbitrarily, and is used to match the experimental setting of Nguyen et al. (2024). To randomize the quadrature and obtain an unbiased estimator, we simply apply a single uniformly drawn rotation to all points. In the two-dimensional setting, we will also include the regular grid on $[-\pi, \pi)$, with a random rotation of uniformly drawn angle $\theta \sim \mathcal{U}[-\pi, \pi)$.

## 4 New candidate estimators

We propose new estimators for the integral inherent to the sliced Wasserstein distance. Our first proposition is a natural importance sampling baseline, and the rest are repulsive methods: three DPPs, a repelled point process. For the DPPs, we select existing DPPs in the probability literature and motivate our selection by applying existing theoretical results to the particular case of the sliced Wasserstein integrand. The novelty there is thus in the application of these DPPs, rather than in the creation of a novel kernel or, say, a new central limit theorem. All estimators will be numerically compared in the experimental section.

### 4.1 An importance sampling baseline

Crude Monte Carlo approximates the integral in (2) using i.i.d. samples from the uniform measure $\mathrm{d}\theta$ on the sphere. Importance sampling consists in rather drawing $\theta_1, \ldots, \theta_N$ from a measure with density $g$ with respect to $\mathrm{d}\theta$, and then to define the estimator

$$I_N^{\mathrm{IS}, g}(f) = \frac{1}{N} \sum_{i=1}^{N} \frac{f(\theta_i)}{g(\theta_i)}. \tag{13}$$

It is unbiased by construction, and the choice of the *proposal distribution* $g$ which minimizes $\mathrm{Var}(I_N^{\mathrm{IS}, g}(f))$ is $g_{\mathrm{opt}} \propto |f|$ (Robert & Casella, 2004) [Theorem 3.12]. Since this proposal is not available in practice, several schemes have been proposed to approximate it using part of one's computational budget in evaluations of the integrand. For instance, limiting ourselves to proposal distributions in the parametric family

$$\mathcal{G}_{\mathrm{vmf}} = \left\{ \frac{1}{2}\mathrm{vmf}(\cdot | \varepsilon, \kappa) + \frac{1}{2}\mathrm{vmf}(- \cdot | \varepsilon, \kappa); \quad \mathrm{vmf}(\cdot | \varepsilon, \kappa) = C(\kappa)\exp(\kappa \varepsilon^T(\cdot)) \quad | \quad \kappa > 0, \, \varepsilon \in \mathbb{S}^{d-1} \right\}$$

of symmetrized von Mises-Fisher distributions, we spend a fixed fraction $r \in (0, 1)$ of our $N$ evaluations of the integrand to find the PDF $g^\star$ in $\mathcal{G}_{\mathrm{vmf}}$ that minimizes an estimate of the KL divergence between $g$ and $g_{\mathrm{opt}}$; this is the so-called cross-entropy method (Kroese et al., 2013); see Appendix A.2 for numerical details on how we perform the fit.

### 4.2 Three determinantal point processes

#### 4.2.1 Orthogonal polynomial ensembles on spherical coordinates.

Representing points on the sphere by their spherical coordinates, we can obtain a DPP on the sphere by mapping a DPP on $\mathbb{X} = [0, 2\pi]^{d-2} \times [0, \pi]$; changes of coordinates are $C^1$-diffeomorphisms and thus preserve DPPs (Lavancier et al., 2015, Proposition A.1.). As a DPP baseline, we thus blindly follow Bardenet & Hardy (2020), who use a projection DPP (Definition 1) with eigenfunctions $(\phi_k)$ in (1) being the products of Legendre polynomials, orthogonal with respect to the uniform distribution. Efficient rejection sampling routines that implement the chain rule (3) are available in the Python package DPPy Gautier et al. (2019b). A central limit theorem for a simple estimator built on such a DPP is available in Bardenet & Hardy (2020), thus potentially helping us obtain asymptotic confidence intervals. However, our integrand (6) is not regular enough, nor is compactly supported within the interior of $\mathbb{X}$ as required in the results of Bardenet & Hardy (2020). Intuitively, we should rather use DPPs that handle both the manifold structure of the sphere and allow for less smooth integrands.

#### 4.2.2 The spherical ensemble.

In the specific setting $d = 3$, another projection DPP over $\mathbb{S}^2$ is available in the probability literature, the so-called *spherical ensemble*. The spherical ensemble comes from random matrix theory, with a dedicated sampling algorithm by construction.

**Definition 3 (Spherical ensemble, Theorem 3 in Krishnapur (2009))** *Let $A$ and $B$ be standard i.i.d. $N \times N$ complex Gaussian matrices. Consider $\pi : \mathbb{S}^2 \setminus \{\mathrm{North}\} \to \mathbb{C}$ the stereographic projection (North being the North pole), and $\lambda_1, \ldots, \lambda_N$ the eigenvalues of the random matrix $A^{-1}B$. Then $\mathcal{S}_N = \{\pi^{-1}(\lambda_1), \ldots, \pi^{-1}(\lambda_N)\}$ is a DPP with respect to the uniform measure $\mathrm{d}\theta$ on the sphere, of almost sure cardinality $N$.*

This point process is naturally repulsive as can be observed on Figure 2.

There are several results that support using the spherical ensemble for Monte Carlo integration on the sphere. Berman (2024) showed that, akin to quasi-Monte Carlo designs, it has low discrepancy with high probability, thus yielding a fast-decaying worst-case integration error for smooth functions. In a more Monte Carlo vein, there exist fast central limit theorems for the spherical ensemble under weak smoothness assumptions.

Indeed, for our integrand (6), Theorem 1 in Rider & Virag (2007) and Theorem 2.5 in Marzo Sánchez et al. (2024) imply

$$\mathrm{Var}\left(\sum_{\theta \in \mathcal{S}_N} f^{(p)}_{\mu,\nu}(\theta)\right) \to \int_{\mathbb{S}^2} \|\nabla f^{(p)}_{\mu,\nu}\|^2 \mathrm{d}\theta. \tag{14}$$

and

$$N\left(\frac{1}{N}\sum_{\theta \in \mathcal{S}_N} f^{(p)}_{\mu,\nu}(\theta) - \int_{\mathbb{S}^2} f^{(p)}_{\mu,\nu}\mathrm{d}\theta\right) \overset{law}{\to} \mathcal{N}\left(0, \int_{\mathbb{S}^2} \|\nabla f^{(p)}_{\mu,\nu}\|^2 \mathrm{d}\theta\right). \tag{15}$$

The only smoothness assumption needed if for the variance in (15) to be finite. In our case, this follows from our integrand being Lipschitz continuous, so that it has an almost-everywhere bounded gradient by Rademacher's theorem (see e.g. Cheeger (1999); although when both measures are discrete, supported on $M$ points, Rademacher's theorem can be replaced by noting that the integrand is $\mathcal{C}^\infty$ except on a finite union of great circles). The estimator in (15) has the fastest converging mean-square error in $d = 3$ among known results for the estimators of the sliced Wasserstein discussed in this paper, beating the rate in $1/N^{1+1/2} = 1/N^{3/2}$ associated to the control variates in (11). We thus expect the spherical ensemble to dominate Monte Carlo estimators in $d = 3$. A major downside of the spherical ensemble is that it is hard to generalize in higher dimensions; see Beltrán & Etayo (2019) and Lemoine & Bardenet (2024). There is however a close cousin to the spherical ensemble that generalizes to any dimension.

### 4.2.3 The harmonic ensemble.

Following the formulation given in Marzo Sánchez et al. (2024) and Beltrán et al. (2016), let $\mathcal{H}_\ell$ be the space of homogeneous harmonic polynomials in $\mathbb{R}^d$ of degree $\ell$, restricted to the sphere $\mathbb{S}^{d-1}$, and $h_\ell = \dim(\mathcal{H}_\ell)$. The harmonic ensemble is the DPP with respect to the uniform measure on the sphere and with kernel

$$K(x,y) = \frac{\pi_L}{\binom{L+(d-1)/2}{L}} P_L^{((d-1)/2,(d-1)/2-1)}(x^T y), \tag{16}$$

where $\pi_L = h_0 + \cdots + h_L$ and $P_L^{((d-1)/2,(d-1)/2-1)}$ is a Jacobi polynomial (Gautschi, 2004). One can show that it is a projection DPP in the sense of Definition 1, where the eigenfunctions $(\phi_k)$ are given by spherical harmonics $\{Y_k^\ell, \ell \geq 0, 1 \leq k \leq h_\ell\}$; see Appendix A.1. The harmonic ensemble can be sampled using the chain rule (3), although for $d = 2$, there is a simpler random matrix model, as the harmonic ensemble is known in this particular case as the Circular Unitary Ensemble (CUE), which is the law of the eigenvalues of a Haar-distributed unitary matrix; see e.g. Remark 4.1.7 in Anderson et al. (2010). A realization of this specific point process on $\mathbb{S}^2$ can be observed in Figure 2.

Like the spherical ensemble, a strong motivation for using the harmonic ensemble is the availability of a fast central limit theorem that translates into small asymptotic confidence intervals for Monte Carlo integration. Indeed, letting $\mathcal{S}_N = \{\theta_1, \ldots, \theta_N\}$ be the harmonic ensemble, Theorem 2.2 in Marzo Sánchez et al. (2024) implies that for our integrand (6),

$$\lim_{N\to\infty} \frac{1}{N^{1-\frac{1}{d-1}}} \mathrm{Var}\left[\sum_{\theta \in \mathcal{S}_N} f^{(p)}_{\mu,\nu}(\theta)\right] = \left\|\left| f^{(p)}_{\mu,\nu}\right|\right\|_{\frac{1}{2}}^2.$$

Moreover,

$$\sqrt{N^{1+\frac{1}{d-1}}}\left(\frac{1}{N}\sum_{\theta \in \mathcal{S}_N} f^{(p)}_{\mu,\nu}(\theta) - \int_{\mathbb{S}^d} f^{(p)}_{\mu,\nu}(\theta)\mathrm{d}\theta\right) \overset{law}{\to} \mathcal{N}(0, \left\|\left| f^{(p)}_{\mu,\nu}\right|\right\|_{\frac{1}{2}}^2).$$

where $\|\cdot\|_{\frac{1}{2}}$ is a specific semi-norm on the Sobolev space $H^{\frac{1}{2}}(\mathbb{S}^{d-1})$ that is equivalent to the semi-norm

$$[f]_{\frac{1}{2}} := \iint_{\mathbb{S}^{d-1}\times\mathbb{S}^{d-1}} \frac{|f(x)-f(y)|^2}{\eta(x,y)^d}\mathrm{d}x\mathrm{d}y, \quad f \in L^2, \tag{17}$$

where $\eta(\cdot,\cdot)$ is the geodesic distance on $\mathbb{S}^{d-1}$; see Marzo Sánchez et al. (2024). To wit, $H^{\frac{1}{2}}(\mathbb{S}^{d-1})$ is the function space for which this quantity is finite. The Lipschitz continuity of $f^{(p)}_{\mu,\nu}$ ensures that $f^{(p)}_{\mu,\nu} \in H^{\frac{1}{2}}(\mathbb{S}^{d-1})$,

so that Marzo Sánchez et al. (2024) [Theorem 2.2] applies and gives the aforementioned central limit theorem.

Note that this definition of the harmonic ensemble constrains us to sample a specific number of points, $\pi_L$. It can be interesting to look at what happens in intermediary levels i.e. to consider incomplete harmonic ensembles. This has been implemented but the runtime becomes quite large when the number of points grows.

### 4.3 Repelled point processes on the sphere

To further reduce the computational cost of repulsive Monte Carlo compared to DPPs, quadratic-time alternatives have been considered for integration on $\mathbb{R}^d$, such as the repelled Poisson process of Hawat et al. (2023) that we recall in Section 2.3. We propose a straightforward adaption to the sphere. More precisely, let $\mathbf{X}$ be a finite point configuration on the sphere $\mathbb{S}^{d-1}$, and $x \in \mathbf{X}$. Define

$$F_{s,\mathbf{X}}(x) = \sum_{y \in \mathbf{X},\, y \neq x} \frac{x - y}{\|x - y\|^s}, \tag{18}$$

which we think of as a repulsive force exerted on $x$ by the other points of the configuration. Like Hawat et al. (2023), unless otherwise specified, we take $s = d$. We consider the repelled configuration

$$\tilde{\Pi}_{\epsilon,s}\mathbf{X} = \left\{ \frac{x + \epsilon F_{s,\mathbf{X}}(x)}{\|x + \epsilon F_{s,\mathbf{X}}(x)\|} \mid x \in \mathbf{X} \right\}, \tag{19}$$

where, unlike Hawat et al. (2023), we need to project back onto the sphere. Letting the original configuration be a Poisson point process of intensity $\rho > 0$, tentatively extending the results of Hawat et al. (2023), we expect the estimator

$$\hat{I}^{\mathrm{rep}}_{\tilde{\Pi}_{\epsilon,d}\mathbf{X}}(f^{(p)}_{\mu,\nu}) = \frac{1}{\rho} \sum_{x \in \tilde{\Pi}_{\epsilon,d}\mathbf{X}} f^{(p)}_{\mu\nu}(x). \tag{20}$$

to be an unbiased estimator of the sliced Wasserstein distance between $\mu$ and $\nu$, with reduced variance compared to a sum over $\mathbf{X}$, at least for $\epsilon > 0$ small enough. Similarly, we expect the same properties to hold if the initial point process is a set of $N$ i.i.d. draws from the uniform measure on the sphere.

Note that in Hawat et al. (2023), a choice of $\epsilon$ independent of $f$, and proportional to $\rho^{-1}$ is suggested. Our empirical findings (see A.5) suggest that this should be the correct magnitude for our $\epsilon$ in the case $s = d$. Note also that the whole procedure only requires the computation of all the pairwise distances and hence runs in $\mathcal{O}(N^2)$, as it is the case in the Euclidean setting, where $N$ is the number of projection directions to be sampled. Overall, we mainly focus our study to a binomial point process $\mathbf{X}$ with $N$ points. It is also possible to apply this repelling step to all the other methods presented here. This leads in various cases to a significant variance decrease at a relatively cheap computational cost as we will experimentally show.

## 5    On the variance of the *UnifOrtho* estimator

*UnifOrtho*, as introduced by Rowland et al. (2019) and recalled in Section 3.2.2, is recommended by the recent (Sisouk et al., 2025) for SW estimation in large dimensions. Anticipating on our own experimental results in Section 6, we will recommend it as well. However, a theoretical understanding of the variance of the *UnifOrtho* estimator is lacking, and its proponents even identified cases where the variance might exceed that of a crude Monte Carlo estimator based on i.i.d. samples (Rowland et al., 2019). We contribute here a new derivation for the variance of the *UnifOrtho* estimator, which sheds light on integrands for which it brings variance reduction. This behavior comes from fundamental properties of the spherical harmonics.

For the sake of clarity, recall from Section 3.2 that the spherical harmonics are defined as follows. For $\ell \geq 1$, let $\mathcal{H}_\ell$ be the set of harmonic homogeneous polynomials of degree $\ell$ restricted to $\mathbb{S}^{d-1}$, and $h_\ell$ its dimension. Take $\mathsf{Y}_0^0 = 1$ to be constant. Then, for $\ell \geq 1$, take $\mathsf{Y}_k^\ell$, $1 \leq k \leq h_\ell$, to be any orthonormal basis of $\mathcal{H}_\ell$. The family of all $\mathsf{Y}_k^\ell$, $\ell \geq 0$, $1 \leq k \leq h_\ell$, is an orthonormal basis of $L^2(\mathbb{S}^{d-1})$. For more details about spherical harmonics, see Appendix A.1 or (Dai & Xu, 2013).

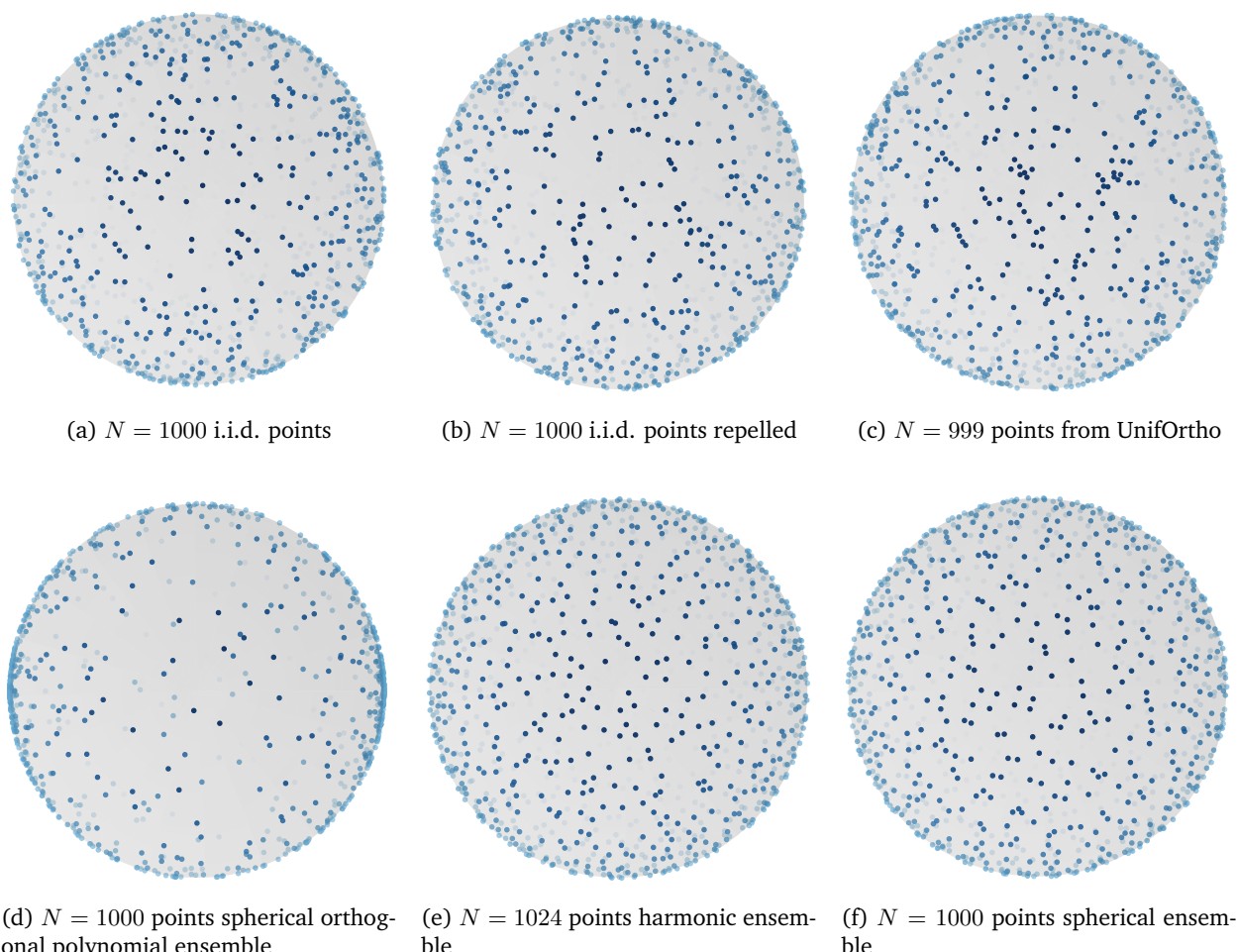

(a) $N = 1000$ i.i.d. points

(b) $N = 1000$ i.i.d. points repelled

(c) $N = 999$ points from UnifOrtho

(d) $N = 1000$ points spherical orthogonal polynomial ensemble

(e) $N = 1024$ points harmonic ensemble

(f) $N = 1000$ points spherical ensemble

Figure 2: Various point processes over the sphere

**Proposition 4** *Let $f$ be a continuous function on $\mathbb{S}^{d-1}$, and $(X_1|\ldots|X_d)$ be a matrix drawn from the Haar measure on the orthogonal group $O(d)$. Let $\hat{f}(\ell, k) = \int_{\mathbb{S}^{d-1}} f(x) Y_k^\ell(x) \mathrm{d}x$ denote the spherical coefficients of $f$. Then*

$$\mathrm{Var}\left(\frac{1}{d}\sum_{i=1}^{d} f(X_i)\right) = \frac{1}{d}\mathrm{Var}(f(X_1)) - \frac{d-1}{d}\sum_{\ell=1}^{+\infty}(-1)^{\ell-1}\lambda_{2\ell}\mu_{2\ell}(f) \tag{21}$$

$$= \frac{1}{d}\mathrm{Var}(f(X_1)) - \frac{d-1}{d}\sum_{\ell=1}^{+\infty}\lambda_{4\ell-2}(\mu_{4\ell-2}(f) - \alpha_{2\ell-1}\mu_{4\ell}(f)), \tag{22}$$

*where $\mu_\ell(f) = \sum_{k=1}^{h_\ell}\hat{f}(\ell,k)^2$, $\alpha_\ell = \dfrac{2\ell+1}{2\ell+d-1}$, and $\lambda_{2\ell} = \dfrac{\Gamma(\frac{d-1}{2})\Gamma(\frac{2\ell+1}{2})}{\sqrt{\pi}\Gamma(\frac{2\ell+d-1}{2})}$.*

Note that in spite of the freedom to choose any basis of $\mathcal{H}_\ell$ when defining the spherical harmonics, the coefficients $\mu_\ell(f)$ are well-defined, as they are independent of the choice of these bases.

**Proof:** Expanding the variance and using the invariance by rotation of the Haar measure yields

$$\mathrm{Var}\left(\frac{1}{d}\sum_{i=1}^{d} f(X_i)\right) = \frac{1}{d}\mathbb{E}[f^2(X_1)] + \frac{d-1}{d}\mathbb{E}[f(X_1)f(X_2)] - \mathbb{E}[f(X_1)]^2.$$

By construction of the Haar measure, conditionally on $X_1$, $X_2$ follows the uniform measure $\sigma$ on $\mathbb{S}^{d-1} \cap X_1^\perp$ (i.e., the $d-2$-dimensional Hausdorff measure $\mathcal{H}^{d-2}$, normalized to have mass 1) (Meckes, 2019, chapter 1.2). In particular,

$$\mathbb{E}[f(X_1)f(X_2)] = \mathbb{E}[f(X_1)\mathbb{E}[f(X_2)|X_1]] = \mathbb{E}[f(X_1)\mathcal{F}f(X_1)], \tag{23}$$

where $\mathcal{F}f(u) = \int_{\mathbb{S}^{d-1}\cap u^\perp} f(w)\mathrm{d}\sigma(w)$ is the Funk transform of $f$. Now, combining Theorem 3.4 and Example 3.12 in Rubin (2024) shows that the spherical harmonics are eigenvectors of the Funk transform. More precisely, for all $\ell \in \mathbb{N}$ and $1 \le k \le h_\ell$, $\mathcal{F}\mathsf{Y}_k^{2\ell+1} = 0$ and

$$\mathcal{F}\mathsf{Y}_k^{2\ell} = (-1)^\ell \lambda_{2\ell}\mathsf{Y}_k^{2\ell}.$$

We note in passing that this is analogous to the classical Funk-Hecke formula Dai & Xu (2013)[Theorem 2.9] and comes from the reproducing property of the spherical harmonics kernel

$$\mathsf{Z}_\ell(x,y) = \sum_{k=1}^{h_\ell} \mathsf{Y}_k^\ell(x)\mathsf{Y}_k^\ell(y)$$

for $\ell \ge 1$. Finally, decomposing $f$ as $f = \sum_{\ell=0}^{\infty}\sum_{k=1}^{h_\ell} \hat{f}(\ell,k)\mathsf{Y}_k^\ell$ and reporting into (23) yields

$$\mathbb{E}[f(X_1)f(X_2)] = \sum_{\ell=0}^{+\infty}(-1)^\ell \lambda_{2\ell}\mu_{2\ell}(f).$$

Now $\mu_0(f) = \mathbb{E}[f(X_1)]^2$, $\lambda_0 = 1$, and standard properties of the Gamma function show that $\lambda_{2\ell} = \alpha_\ell\lambda_{2\ell-2}$. Combining these facts gives the result. $\qquad\square$

Proposition 4 calls for comments. The first term in both (21) and (22) is the variance of the crude Monte Carlo estimator, and (21) and (22) are two different expressions for the difference in variance between *UnifOrtho* and that crude Monte Carlo estimator. First, it is clear from e.g. (21) that one can get either a decrease or an increase in variance from *UnifOrtho*, depending on the "energy profile" $(\mu_{2\ell}(f))_{\ell\in\mathbb{N}}$ of the integrand $f$. This explains the observed increase in variance in an example of Rowland et al. (2019). To make another more extreme example, note that $\lambda_2 = 1/d - 1$, so that

$$\mathrm{Var}\left(\frac{1}{d}\sum_{i=1}^{d}\mathsf{Y}_k^2(X_i)\right) = 0$$

for all $k$. In contrast, integrating $\mathsf{Y}_k^4$ leads to an increase in variance compared to crude Monte Carlo. Second, we note that the sum in (21) is alternating: each nonpositive term in the sum is followed by a nonnegative term. In $d = 2$, $\lambda_{2\ell} = 1$ for all $\ell$, so that each term carries the same weight, and a single large isolated $\mu_{2\ell}$ at some high even frequency $\ell$ can be responsible for a variance increase in (21). When the dimension grows, the generalized Stirling formula yields $\lambda_{2\ell} = \mathcal{O}(\ell^{-\frac{d-2}{2}})$, so that only the first terms in either (21) or (22) carry significant weight. The interest of (22) is to show the effect of dimension growth on a sequence of integrands with the same spectral profile $(\mu_{2\ell}(f))$ throughout dimensions: as $\alpha_\ell$ for a fixed $\ell$ decreases as $1/d$, nonnegative terms get attenuated more and more, and the variance overall decreases. Third, note that the Funk transform sends all odd-degree spherical harmonics to zero, and in particular *UnifOrtho* has the same variance as crude Monte Carlo for an odd integrand. The integrand (6) in the sliced Wasserstein distance is even, hence only decomposes onto even harmonics, in coherence with *UnifOrtho*'s success.

## 6 Experiments

In this section, we numerically illustrate the repulsive Monte Carlo estimators of Section 4. The methods we compare are often referred to using acronyms.

- *i.i.d.* is classical Monte Carlo with i.i.d. uniform points on the sphere; it is the default baseline.

- *ISVMF* is short for importance sampling with von-Mises Fischer proposal; see Section 4.1.

- *UnifOrtho* refers to the union of independent Haar-distributed bases introduced in Rowland et al. (2019); see Sections 3.2.2 and 5.

- *CV up* (resp. *CV low*) is short for Control Variates "up" (resp. "low") as in Nguyen & Ho (2024); see Section 3.2.1, Equation 10 (resp. 9).

- *SHCV* is short for Spherical Harmonics control variates Sliced Wasserstein, as introduced by Leluc et al. (2024) and described in Section 3.2.1, Section 3.2.1.

- *Repelled* is described in Section 4.3, specifically in (19), while *Repelled SHCV* corresponds to using spherical harmonics control variates (see Section 3.2.1) built on the repelled points. To be more precise, we first sample points according to the repelled point process from (19), to which we then apply the SHCV process from Section 3.2.1.

- The three DPPs from Section 4.2 are denoted as *OPE* for the stereographic projection of the multivariate Jacobi orthogonal polynomial ensemble from Section 4.2.1, *Harmonic* for the harmonic ensemble from Section 4.2.3, and *Spherical* for the spherical ensemble from Section 4.2.2. Note that *CUE* (short for Circular Unitary Ensemble), is the 2-dimensional version of the harmonic Ensemble.

- *Spherical SHCV*, only present when $d = 3$, consists in applying spherical harmonics control variates to the spherical ensemble.

- Finally, *QMC* or *Randomized regular grid* corresponds to the randomized quasi-Monte Carlo grids in $d \in \{2, 3\}$ described in Section 3.2.2 in Equation 12.

Following Sections 3.2, 4 and 5, we expect to witness the following decay rates. If $N$ is the number of projections considered, the classical variance decay rate of *i.i.d.* points is $1/N$. Some methods enjoy faster than *i.i.d.* decay rate, such as the *harmonic ensemble* from Section 4.2.3 or the *SHCV* method from Section 3.2.1 at $1/N^{1+1/(d-1)}$. In the specific case of $d = 3$, the fastest known method is the *spherical ensemble*, from Section 4.2.2 at a rate $1/N^{1+2/2} = 1/N^2$. We cannot say much about the decay rate of the other methods, except from *UnifOrtho*, which is still at $1/N$ but with a different constant as stated in Section 5.

We consider three different experimental settings. The first one is a toy example where we compute the SW distance between two independent Gaussian samples. In order to see how our algorithms behave when comparing more realistic point clouds, we then compute the $SW_2$ distance between pairs of datasets from a database of three-dimensional point clouds (Chang et al., 2015) used in previous papers on the SW distance (Leluc et al., 2024; Nguyen & Ho, 2024). Finally, to generate a different kind of realistic point clouds, we place ourselves in the position of a researcher who wants to compare the outputs of various MCMC algorithms, a task for which the SW has recently been used (Cardoso et al., 2023; Linhart et al., 2024). This time, we focus on $SW_1$ rather than $SW_2$, since the former corresponds to a worst-case integration error, a natural figure of merit to compare MCMC algorithms.

The implementations used are available here https://github.com/vladpetrovic/SWMC.

## 6.1 Gaussian toy example

For any given dimension $d$, we sample two independent vectors $m_X, m_Y$ from $\mathcal{N}(0, I_d)$, and, independently, two matrices $U, V$ from $\mathcal{N}(0, I_{d \times d})$. Consider then $\Sigma_X = U^T U$, and $\Sigma_Y = V^T V$. Finally, sample $x_1, \ldots, x_M$ (resp. $y_1, \ldots, y_M$) i.i.d. from $\mathcal{N}(m_X, \Sigma_X)$ (resp $\mathcal{N}(m_Y, \Sigma_Y)$), and define

$$\mu = \frac{1}{M} \sum_{i=1}^{M} \delta_{x_i}, \quad \nu = \frac{1}{M} \sum_{i=1}^{M} \delta_{y_i}.$$

For each dimension and number of projections, we consider 100 independent realizations of each estimator. In $d = 2$ and $d = 10$, for SHCV, a maximal degree of 4 for the spherical harmonics is fixed, as in the original

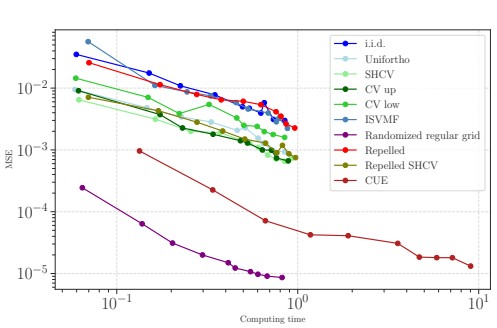
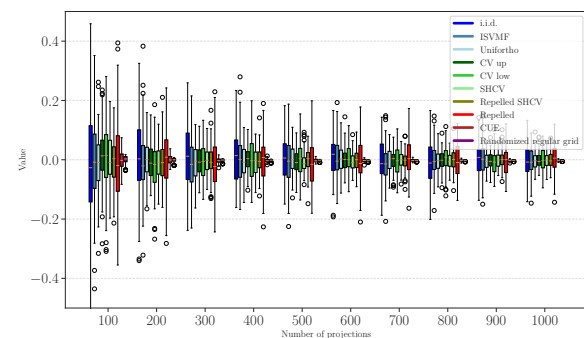

(a) MSE vs computing time for Gaussian toy example, $d = 2$

(b) Errors vs number of projections for Gaussian toy example, $d = 2$

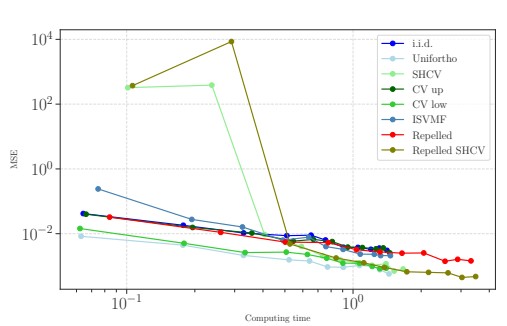
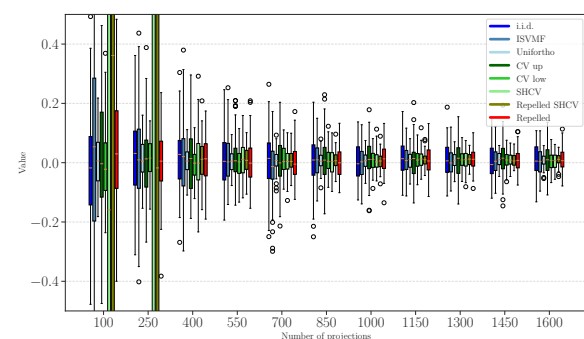

(c) MSE vs computing time for Gaussian toy example, $d = 10$

(d) Errors vs number of projections for Gaussian toy example, $d = 10$

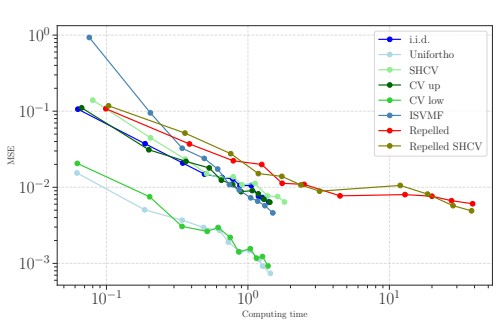
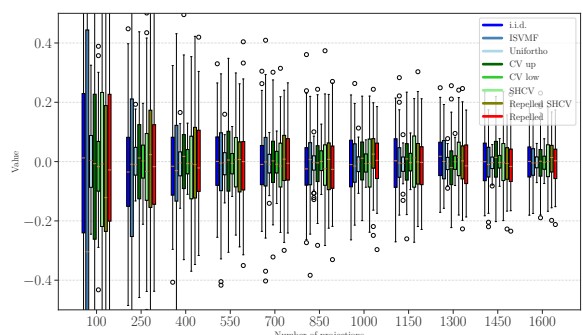

(e) MSE vs computing time for Gaussian toy example, $d = 20$

(f) Errors vs number of projections for Gaussian toy example, $d = 20$

Figure 3: Results for the Gaussian toy example, across $d = 2, 10, 20$. The actual value of the 2-sliced Wasserstein distance is estimated using Monte Carlo integration with $10^6$ projections.

paper (Leluc et al., 2024). For $d = 20$, this maximal degree is reduced to $2$. Empirically, up to $1600$ projection points in $d = 20$, the number of control variates corresponding to a maximal degree of $3$ is indeed too large for the estimator to get near the (known) value of sliced Wasserstein. Note that a similar phenomenon is

observed in $d = 10$ on 100 projections or 250 projections (see Figure 3d). This is related to the requirement fixed in Equation 11 for the estimator to be consistent.

The results are given in Figure 3, with the left panel showing estimated mean-squared errors vs. computing time, and the right panel showing boxplots of the integration errors. The reference values are computed with a comparatively long Monte Carlo run.

For $d = 2$, Figure 3a highlights that the randomized regular grid far outperforms any other method in terms of MSE. The determinantal point process CUE stands as second, and all the other methods stand in the same range in terms of MSE. Things are different in the 10- and 20- dimensional settings, where the randomized grid and the DPPs do not feature anymore among the leading methods. In $d = 10$, as per Figures 3c and 3d, the differences between the methods are less sharp, but *UnifOrtho* dominates, closely followed by *CV low*, as well as *SHCV* and *Repelled SHCV*, once there are enough projections for the linear systems for consistency to show. In $d = 20$ dimensions, the only relevant methods seem to be *UnifOrtho* and *CV low*, which far outperform any other method. These conclusions are coherent with the ones presented in Sisouk et al. (2025).

Overall, repulsive methods are among the leading methods in each dimension, but no single repulsive method uniformly dominates: as expected, a randomized grid or a well-chosen DPP are adequate in low dimension, while higher dimensions seem to favor *UnifOrtho*. Maybe surprisingly, we note that repelling the points seems to have only a moderate effect on the MSE. This effect is not even guaranteed to be a decrease in the MSE, and we will investigate this more quantitatively in the Appendix A.5.

### 6.2 Three-dimensional point clouds

We now consider three-dimensional point clouds Chang et al. (2015) in the Shapenet database. They are configurations of points that cover shapes that range from simple cylinders to planes or benches. We arbitrarily consider four point clouds from the database, and compute the difference between point clouds #2 and #34, and the distance between point clouds #3 and #35. The point clouds are shown in Figure 4.

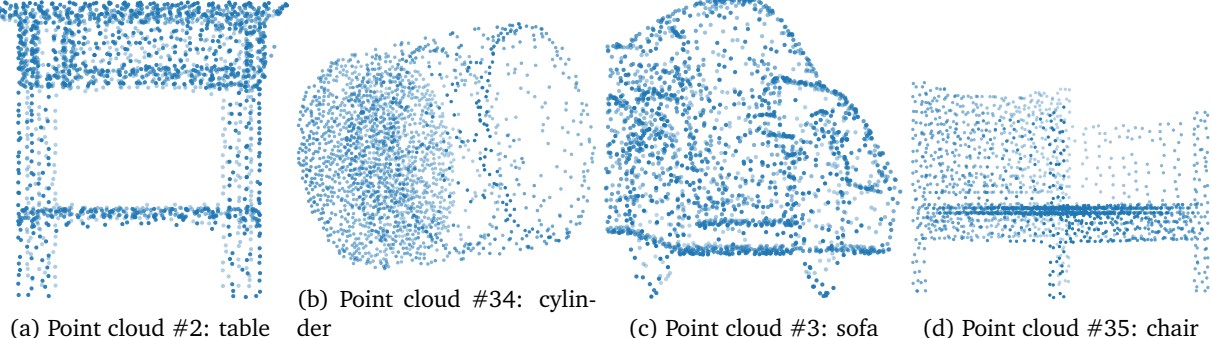

(a) Point cloud #2: table     (b) Point cloud #34: cylinder     (c) Point cloud #3: sofa     (d) Point cloud #35: chair

Figure 4: Two-dimensional projections of the various point clouds used in Section 6.2.

The typical integrand in the $SW_2$ between two such point configurations looks quite different from the toy Gaussian case of Section 6.1. In particular, it can be multimodal; see Figure 9. The results of our experiments are shown in Figure 5.

The results are comparable with those of the two-dimensional Gaussian toy example of Section 6.1, comforting the conclusion that in dimension $d \leq 3$, for the integrands and the regimes we consider, randomized grids should be the default quadrature: they are both cheap to sample and provide significantly more accurate integral estimators than sophisticated Monte Carlo methods such as *SHCV* or DPPs like the spherical ensemble.

Among the other methods, three seem to be of similar performance: *SHCV*, repelled *SHCV* and the spherical ensemble. As the number of projection directions grows, the spherical ensemble gains an edge over the other two methods, in accordance to its faster variance decay (15). A further improvement is obtained by

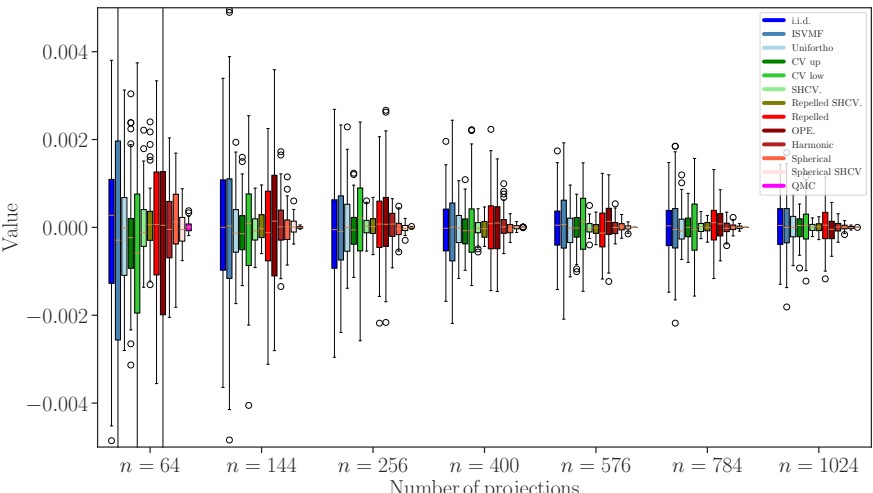

(a) Errors for the $SW_2$ between point clouds #2 and #34.

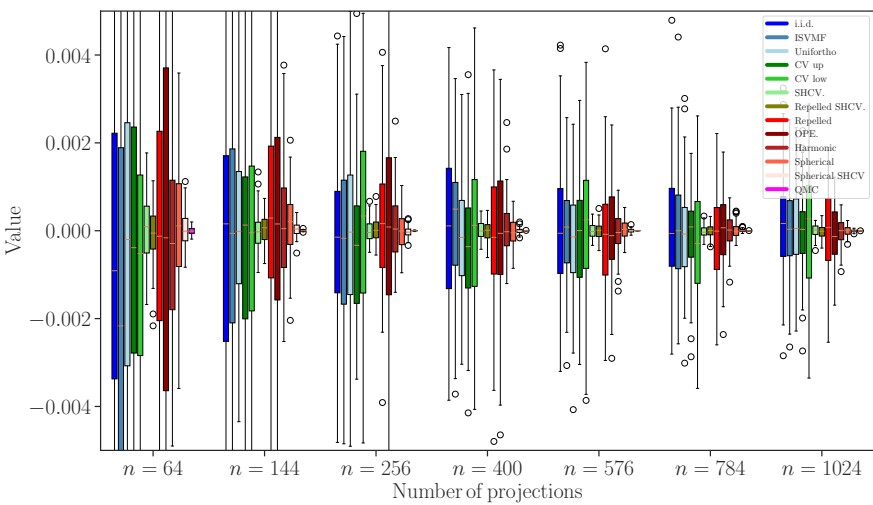

(b) Errors for the $SW_2$ between point clouds #3 and #35.

Figure 5: Boxplots of the errors for three-dimensional point clouds. The boxplots are centered around a reference value of the sliced Wasserstein estimated using QMC with $10^5$ points.

combining the spherical ensemble with *SHCV*, i.e. evaluating the *SHCV* estimator on a spherical ensemble realization rather than i.i.d. points. Finally, we note that *ISVMF* does not necessarily reduce the MSE of the i.i.d. estimator: this is likely due to the multimodality of the integrand, which is not reflected in the von Mises-Fischer proposal.

To understand the behavior of the *UnifOrtho* estimator in the specific case, we used a QMC sequence to estimate the spectral profile $(\mu_{2\ell}(f))$, where $f$ is the integrand of the sliced Wasserstein between two point cloud, we refer to Proposition 4 for further details. Figure 6 shows in both cases a fast decay of these coefficients, with a sharper slope for comparing point clouds #2 and #34. This explains the higher gain of *UnifOrtho* in this case, as seen in Figure 5.

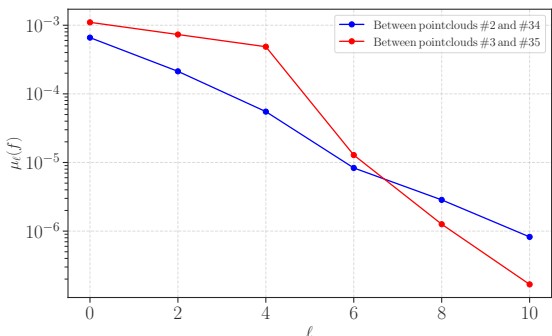

Figure 6: Evolution of $\mu_\ell$ for the two integrands appearing in $SW_2$ in Section 6.2.

## 6.3 Comparing MCMC kernels

To provide a realistic use case of the SW distance in high and arbitrary dimension, we consider the numerical validation of an MCMC kernel. Numerically assessing that an MCMC kernel targets the expected distribution, as well as comparing MCMC kernels in terms of integration errors with respect to a target distribution, are natural tasks in computational statistics and machine learning. In that context, the sliced Wasserstein between a realization of an MCMC history and a known target distribution can be used as a figure of merit, as done e.g. in Cardoso et al. (2023). To see why, note first that the 1-Wasserstein distance is a worst-case integration error. Indeed, the classical dual formulation of the $W_1$ distance reads

$$W_1(\mu, \nu) = \sup_{f: \mathrm{Lip}(f) \leq 1} \left| \int f \mathrm{d}\mu - \int f \mathrm{d}\nu \right|, \tag{24}$$

where $\mathrm{Lip}(f)$ is the Lipschitz constant of the (Lipschitz) function $f$; see e.g. (Peyré & Cuturi, 2018). Second, $SW_1$ can be used as a proxy for $W_1$, in the sense of the equivalence in (5). Hence, the law of the $SW_1$ distance between a (random) MCMC history and the target distribution provides information on the integration error incurred by the MCMC kernel.

More formally, assume that the MCMC algorithm targets a distribution $\mu$, and outputs a random configuration of points $(X_1, \ldots, X_T)$. Call

$$\mu_T^{\mathrm{MCMC}} = \frac{1}{T} \sum_{i=1}^T \delta_{X_i}$$

the corresponding (random) empirical measure. Note that evaluating directly $SW_1(\mu_T^{\mathrm{MCMC}}, \mu)$ is not possible, but if it is possible to sample $Y_1, \ldots, Y_M$ i.i.d. from $\mu$ (as is often the case when testing sampling algorithms on simple targets), we can use

$$\mu_M^{\mathrm{iid}} = \frac{1}{M} \sum_{i=1}^M \delta_{Y_i}$$

as a proxy for $\mu$. The triangular identity indeed guarantees

$$SW_1(\mu_T^{\mathrm{MCMC}}, \mu) \leq SW_1(\mu_T^{\mathrm{MCMC}}, \mu_M^{\mathrm{iid}}) + SW_1(\mu_M^{\mathrm{iid}}, \mu). \tag{25}$$

The second term of the right-hand side can be controlled via results involving the sample complexity (Manole et al., 2022), and scales as $1/\sqrt{M}$. We thus focus on estimating $SW_1(\mu_T^{\mathrm{MCMC}}, \mu_M^{\mathrm{iid}})$, using Monte Carlo integration over the sphere. Note that, since our goal is to illustrate various quadratures on the sphere, we will consider a single realization of $\mu_T^{\mathrm{MCMC}}$ per dimension, but an MCMC practitioner wanting to estimate the quality of an MCMC kernel should repeatedly sample independent MCMC histories and consider the distribution of the obtained SW distances.

We consider $d \in \{10, 30\}$. Our target distribution is the banana-shaped target that is classically used to demonstrate the ability of gradient-based MCMC samplers, such as Hamiltonian Monte Carlo (Duane et al., 1987), to make long-range jumps and thus reduce the asymptotic variance of the corresponding MCMC estimators. Formally, the banana-shaped target is the distribution of the image of a Gaussian vector $X \sim \mathcal{N}(0, I_d)$ by the map $f : \mathbb{R}^d \to \mathbb{R}^d$ defined by $f_{2j+1}(X) = x_{2j+1}$ and $f_{2j+2}(X) = -x_{2j+2} + (x_{2j+1} - 5)^2$ for $j \geq 0$. We further fix the number of projections in the SW estimators to $N = 10^3$, the number of points on which the reference measure is supported to $M = 10^4$.

To obtain realizations of $\mu_T^{\mathrm{MCMC}}$, we consider four MCMC kernels from the *PyMC* v5.23.0. library (Abril-Pla et al., 2023), namely four variants of Hamiltonian Monte Carlo (HMC; (Duane et al., 1987)). HMC has several hyperparameters, such as a *stepsize* and a *mass matrix* parameters, and *PyMC* offers different options to tune them. Our first kernel (henceforth referred to as *regular HMC*) is the default automatic tuning in *PyMC*. Our second kernel (*broken HMC*) corresponds to us manually blocking the adaptation of the mass matrix, and setting it to the identity matrix. Our third kernel (*regular NUTS*) is the No-U-Turn adaptive HMC sampler of Hoffman et al. (2014), as implemented again in *PyMC*. Our fourth kernel (*broken NUTS*) is NUTS, but with us manually blocking the online adaptation of the stepsize parameter –which renders the analysis of the Markov chain difficult– and setting it to a fixed value. The objective it to observe the practical relevance of the hyperparameter tuning mechanisms in HMC.

In $d = 10$, we consider five estimators of the $SW_1$ distance, namely *i.i.d.*, *UnifOrtho*, *Repelled i.i.d.*, *SHCV*, and *Repelled SHCV*. For *SHCV*, we were able to set the maximum degree of spherical harmonics to 4. In $d = 30$, we keep *i.i.d.*, *UnifOrtho*, and *Repelled i.i.d.*. Note that we do not consider *CV up* and *CV low*, since they were specifically designed for $SW_2$.

Our results for $d = 10, 30$ are respectively shown in Figures 7 and 8. We show the average of $1,000$ independent realizations of each estimator, with the $95\%$ Gaussian confidence interval for the mean, Bonferroni corrected across the $5 \times 4$ estimators (respectively $3 \times 4$) corresponding to each value of $T$. Strictly speaking, one can thus statistically compare all confidence intervals for any given value of $T$, but we should refrain from comparing across values of $T$. Since our objective is to compare the accuracies of various quadratures, this seemed a natural correction.

| $T$ | i.i.d. | Repelled | UnifOrtho | SHCV | Repelled SHCV |
|---|---|---|---|---|---|
| | $4.955 \pm 0.093$ | $4.957 \pm 0.062$ | $\mathbf{4.958 \pm 0.028}$ | $4.960 \pm 0.049$ | $4.957 \pm 0.037$ |
| 10 | $4.959 \pm 0.101$ | $4.960 \pm 0.068$ | $\mathbf{4.958 \pm 0.026}$ | $4.957 \pm 0.052$ | $4.957 \pm 0.040$ |
| | $4.957 \pm 0.089$ | $4.957 \pm 0.073$ | $\mathbf{4.958 \pm 0.026}$ | $4.956 \pm 0.060$ | $4.959 \pm 0.039$ |
| | $4.805 \pm 0.102$ | $4.811 \pm 0.062$ | $\mathbf{4.811 \pm 0.023}$ | $4.812 \pm 0.050$ | $4.812 \pm 0.041$ |
| | $0.741 \pm 0.007$ | $0.741 \pm 0.005$ | $\mathbf{0.741 \pm 0.002}$ | $0.741 \pm 0.004$ | $0.741 \pm 0.003$ |
| 100 | $0.847 \pm 0.012$ | $0.848 \pm 0.008$ | $\mathbf{0.848 \pm 0.003}$ | $0.847 \pm 0.007$ | $0.848 \pm 0.006$ |
| | $4.051 \pm 0.082$ | $4.050 \pm 0.054$ | $\mathbf{4.051 \pm 0.020}$ | $4.049 \pm 0.044$ | $4.051 \pm 0.029$ |
| | $0.579 \pm 0.008$ | $0.579 \pm 0.006$ | $\mathbf{0.580 \pm 0.002}$ | $0.580 \pm 0.005$ | $0.579 \pm 0.004$ |
| | $0.270 \pm 0.003$ | $0.270 \pm 0.002$ | $\mathbf{0.270 \pm 0.001}$ | $0.270 \pm 0.003$ | $0.270 \pm 0.002$ |
| 1 000 | $0.487 \pm 0.009$ | $0.487 \pm 0.006$ | $\mathbf{0.487 \pm 0.002}$ | $0.487 \pm 0.005$ | $0.487 \pm 0.004$ |
| | $0.374 \pm 0.004$ | $0.374 \pm 0.003$ | $\mathbf{0.374 \pm 0.002}$ | $0.374 \pm 0.005$ | $0.374 \pm 0.004$ |
| | $0.242 \pm 0.004$ | $0.242 \pm 0.003$ | $\mathbf{0.242 \pm 0.001}$ | $0.242 \pm 0.003$ | $0.242 \pm 0.002$ |
| | $0.158 \pm 0.002$ | $0.1580 \pm 0.001$ | $\mathbf{0.1580 \pm 8 \cdot 10^{-4}}$ | $0.158 \pm 0.002$ | $0.1580 \pm 0.001$ |
| 10 000 | $0.097 \pm 0.001$ | $0.0968 \pm 7 \cdot 10^{-4}$ | $\mathbf{0.0968 \pm 4 \cdot 10^{-4}}$ | $0.097 \pm 0.001$ | $0.0968 \pm 8 \cdot 10^{-4}$ |
| | $0.790 \pm 0.016$ | $0.790 \pm 0.010$ | $\mathbf{0.790 \pm 0.004}$ | $0.790 \pm 0.008$ | $0.790 \pm 0.006$ |
| | $0.098 \pm 0.001$ | $0.0984 \pm 6 \cdot 10^{-4}$ | $\mathbf{0.0983 \pm 4 \cdot 10^{-4}}$ | $0.0983 \pm 9 \cdot 10^{-4}$ | $0.0983 \pm 6 \cdot 10^{-4}$ |

Figure 7: Averaged $SW_1$ and asymptotic confidence intervals in $d = 10$. The color code is Blue for broken HMC, Red for regular HMC, Green for broken NUTS, and Purple for regular NUTS.

The first conclusion is that *UnifOrtho* consistently yields smaller confidence intervals than the other methods, in both dimensions, which is why we display the corresponding column in bold. Similarly, the second conclusion is that repelled versions of each algorithm reduce the size of the confidence intervals in $d = 10$, but the improvement is less perceptible in $d = 30$. This is to be taken with a pinch of salt, however, as we do not provide a confidence interval on the *variance* of the estimator.

| $T$ | i.i.d. | Repelled | UnifOrtho |
|---|---|---|---|
| 10 | $2.135 \pm 0.017$ | $2.135 \pm 0.017$ | $\mathbf{2.135 \pm 0.002}$ |
| | $1.919 \pm 0.010$ | $1.918 \pm 0.010$ | $\mathbf{1.918 \pm 0.001}$ |
| | $4.057 \pm 0.067$ | $4.063 \pm 0.078$ | $\mathbf{4.063 \pm 0.018}$ |
| | $4.717 \pm 0.089$ | $4.718 \pm 0.086$ | $\mathbf{4.719 \pm 0.022}$ |
| 100 | $0.726 \pm 0.009$ | $0.725 \pm 0.009$ | $\mathbf{0.725 \pm 0.004}$ |
| | $0.849 \pm 0.013$ | $0.848 \pm 0.011$ | $\mathbf{0.848 \pm 0.004}$ |
| | $4.242 \pm 0.077$ | $4.251 \pm 0.079$ | $\mathbf{4.247 \pm 0.020}$ |
| | $0.529 \pm 0.007$ | $0.529 \pm 0.008$ | $\mathbf{0.529 \pm 0.003}$ |
| 1 000 | $0.363 \pm 0.005$ | $0.363 \pm 0.004$ | $\mathbf{0.363 \pm 0.001}$ |
| | $0.288 \pm 0.003$ | $0.288 \pm 0.004$ | $\mathbf{0.288 \pm 0.001}$ |
| | $0.236 \pm 0.003$ | $0.236 \pm 0.003$ | $\mathbf{0.236 \pm 0.001}$ |
| | $0.215 \pm 0.003$ | $0.215 \pm 0.003$ | $\mathbf{0.215 \pm 0.001}$ |
| 10 000 | $0.169 \pm 0.002$ | $0.169 \pm 0.002$ | $\mathbf{0.1693 \pm 6 \cdot 10^{-4}}$ |
| | $0.134 \pm 0.002$ | $0.134 \pm 0.002$ | $\mathbf{0.1341 \pm 6 \cdot 10^{-4}}$ |
| | $0.0645 \pm 8 \cdot 10^{-4}$ | $0.0645 \pm 8 \cdot 10^{-4}$ | $\mathbf{0.0646 \pm 3 \cdot 10^{-4}}$ |
| | $0.0638 \pm 7 \cdot 10^{-4}$ | $0.0638 \pm 8 \cdot 10^{-4}$ | $\mathbf{0.0638 \pm 3 \cdot 10^{-4}}$ |

Figure 8: Averaged $\mathrm{SW}_1$ and asymptotic confidence intervals in $d = 30$. The color code is Blue for broken HMC, Red for regular HMC, Green for broken NUTS, and Purple for regular NUTS.

As to our mock goal to compare algorithms, Figure 7 shows first, for instance, no statistical gain in using NUTS rather than regular HMC when $T = 10,000$. In $d = 30$, a similar phenomenon can be observed in Figure 8 when comparing broken NUTS and regular NUTS when $T = 10,000$. In that case, only *UnifOrtho* has a small enough variance to yield a statistically significant difference in performance, in favor of regular NUTS, as expected. This time, the pinch of salt comes from our use of a single MCMC run for each pair of values of $T$ and $d$. Still, as quadrature algorithms are concerned, *UnifOrtho* is to be preferred.

## 7 Discussion

Our empirical findings suggest that, when working in small dimensions ($d \in \{2, 3\}$), the lowest variance is obtained by randomizing simple deterministic quadratures. Indeed, the randomized spiral points in $d = 3$, and the classical grid in $d = 2$ outperform most sophisticated random methods, at a cheap computational cost. When the dimension grows, these methods become unavailable, and more inherently random quadratures become attractive. Crude Monte Carlo, using *i.i.d.* uniform samples, quickly gets outperformed by most of the presented methods. Among them, DPPs are competitive in smaller dimensions, but their sampling cost becomes prohibitive as dimension increases. This is especially true for the *harmonic ensemble*, whose cardinality is bound to be exponential in the dimension, while sampling intermediate levels requires extensive calls to the spherical harmonics and a rejection sampling phase with a loose rejection bound. On the other hand, the *Repelled* processes are cheap alternatives to DPPs that lead to a (small) variance reduction. Yet, their behavior is less well understood. While Appendix A.5 suggests that some of the intuition on tuning the repulsion carries over from the Euclidean case, combining the repulsion operator with e.g. control variates leads to unstable behavior. Turning to control variates methods, they consistently lead to variance reduction in our benchmark, yet they come with restrictions: *CV up* and *CV low* are limited to $SW_2$, while *SHCV* requires the computation of spherical harmonics. Finally, a clear cost-efficient algorithm outperforms every other method in higher dimension: *UnifOrtho*. This is interesting, as it is a repulsive Monte Carlo estimator, yet with limited negative dependence due to the fact that it is the union of many independent small repulsive point processes. We contributed to the understanding of the success of *UnifOrtho* by providing an expression for the variance the corresponding estimator in terms of the spherical harmonics coefficients of the integrand, which also explains why variance can actually increase if applied to integrands with specific spectral profiles. As stated in our introduction, our observations are independent of the ML task that will use the SW distance, and the important factors affecting the performances of the methods are the dimension of the problem and the number of points on which the measures are supported. Another avenue for future work is to combine *UnifOrtho* and control variates to provide a uniform decrease in variance. Similarly,

understanding the spectral profile of the integrand in the SW distance, in terms of easy-to-estimate features of the two involved distributions, would help choosing the right estimator.

**Acknowledgments**

We acknowledge support from ERC grant BLACKJACK ERC-2019-STG-851866 and ANR AI chair BACCARAT ANR-20-CHIA-0002.

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

# A   Appendix

## A.1   Spherical harmonics

The spherical harmonics are a class of functions that play an important role when approximating functions on the sphere. We refer to Dai & Xu (2013) for a comprehensive introduction, from which we isolate a few points here for completeness.

Let $d \geq 2$, the simplest definition is that the spherical harmonics on $\mathbb{S}^{d-1}$ are the homogeneous harmonic polynomials of $\mathbb{R}^d$, restricted to the sphere $\mathbb{S}^{d-1}$. Alternately, if $\Delta$ is the Laplace-Beltrami operator on $\mathbb{S}^{d-1}$ and $\lambda_\ell = \ell(\ell+d-2)$, the spherical harmonics of order $\ell \in \mathbb{N}$ can be defined as the elements of the eigenspace $\mathcal{H}_\ell$ of $\Delta$ corresponding to eigenvalue $-\lambda_\ell$. One can then show that $\mathcal{H}_\ell$ is the set of harmonic homogeneous polynomials of degree $\ell$ restricted to $\mathbb{S}^{d-1}$, as expected. Furthermore,

$$\Pi_L = \bigoplus_{\ell=0}^{L} \mathcal{H}_\ell$$

is the space of harmonic polynomials in $\mathbb{R}^d$ restricted to $\mathbb{S}^{d-1}$ of degree up to $L$. We also note, following Marzo Sánchez et al. (2024), that

$$\pi_L := \dim(\Pi_L) = \frac{2L + (d-1)}{d-1} \binom{(d-1) + L - 1}{L} = \frac{2}{\Gamma(d)} L^{d-1} + o(L^{d-1}). \tag{26}$$

For a given $\ell$, let $h_\ell = \dim(\mathcal{H}_\ell)$ and $\{\mathsf{Y}_k^\ell | 1 \le k \le h_\ell\}$ be any orthonormal basis of $\mathcal{H}_\ell$. Then Dai & Xu (2013) [Theorem 1.2] state that the elements of $\{\mathsf{Y}_k^\ell | \ell \in \mathbb{N}, 1 \le k \le h_\ell\}$ are centered functions for the uniform measure on the sphere, as soon as $l \ge 1$, which form a Hilbert basis of $L^2(\mathbb{S}^{d-1})$.

From a computational standpoint, it is often useful to note the following addition formula Dai & Xu (2013) [Theorem 2.6],

$$\forall x,\, y \in \mathbb{S}^{d-1},\ \mathsf{Z}_\ell(x,y) := \sum_{k=1}^{h_\ell} \mathsf{Y}_k^\ell(x)\mathsf{Y}_k^\ell(y) = \frac{n+\lambda}{\lambda} C_\ell^\lambda(\langle x,y\rangle), \qquad (27)$$

where $C_\ell^\lambda$ is the Gegenbauer polynomial of degree $\ell$ and $\lambda = \frac{d-2}{2}$. This leads to the following definition.

**Definition 5** *A set of points $\{x_1, \cdots, x_{h_\ell}\} \subset \mathbb{S}^{d-1}$ is said to be fundamental if the matrix $\mathbf{C}_\ell := (C_\ell^\lambda(\langle x_i, x_j\rangle))_{1 \le i,j \le h_\ell}$ is invertible.*

Fundamental sets are particularly interesting since, if $\{x_1, \cdots, x_{h_\ell}\}$ is a fundamental set, then $\{C_\ell^\lambda(\langle \cdot, x_i\rangle) | 1 \le i \le h_\ell\}$ is a basis of $\mathcal{H}_\ell$ (Dai & Xu, 2013) [Theorem 3.3]. This theorem is at the heart of the library[1] developed by Dutordoir et al. (2020) to compute spherical harmonics. Their method consists in greedily building a fundamental set that is likely to lead to a stable Cholesky decomposition $\mathbf{C}_\ell$. This is done by iteratively adding a point that maximizes the determinant of $(C_\ell^\lambda(\langle x_i, x_j\rangle))_{1 \le i,j \le h_\ell}$. Then, through a Cholesky decomposition of $\mathbf{C}_\ell$, they obtain the Gram-Schmidt orthonormalization of $\{C_\ell^\lambda(\langle \cdot, x_i\rangle) | 1 \le i \le h_\ell\}$. In other words, they obtain an orthonormal basis of $\mathcal{H}_\ell$.

The greedy construction of a fundamental set is computationally heavy, although one has to only run it once only. As a computationally cheaper alternative and at the price of stability of the Choleky decomposition, the point sets which are not fundamental lie in $\{\det(\mathbf{C}_\ell) = 0\}$, which is an algebraic hypersurface of $\mathbb{S}^{(d-1) \times h_\ell}$ so is of measure zero. Hence almost every set of points is a fundamental set (Dai & Xu, 2013). However, one cost that cannot be avoided is that all the spherical harmonics of a given level have to be computed, which comes down to finding the Cholesky decomposition of an $h_\ell \times h_\ell$ matrix, and $h_\ell$ grows as $\ell^{d-2}$.

## A.2 More on the importance sampling scheme

For completeness, and because fitting a von-Mises-Fisher distribution is not straightforward, we provide here pseudocode in Algorithm 1 for our fitted importance sampling estimator.

## A.3 Pseudocode for competing methods

For completeness, we add pseudocode to describe the methods we compare to. Algorithm 2 describes the classical HKPV algorithm to sample from a DPP (see Section 4.2), Algorithm 3 describes the SHCV method (see Section 3.2.1), Algorithm 4 describes the crude Monte Carlo with a single control variate algorithm (see Section 3.2.1), Algorithm 5 describes the UnifOrtho sampling strategy (see Section 3.2.2)Note that, the SHCV implementation described here highlights the fact that the weights of the quadrature are independent of the function to integrate.

## A.4 Discussion on the shape of the integrand

We include in Figure 9 various plots of the integrand (6) of the sliced Wasserstein distance in three dimensions. In Figure 9a, we show the integrand corresponding to two Gaussians with random means and covariances, as specified in Section 6.1. In Figure 9b, we examine the integrand (6), but this time between two empirical measures based on respective i.i.d. draws from the same two Gaussians. The integrands in Figures 9a and 9b are visually similar, as expected. Moreover, they seem to be unimodal up to symmetry. In particular, we expect importance sampling with a fitted symmetrized vMF proposal to yield low variance.

Figures 9c and 9d show the integrand (6) for the point clouds used in Section 6.2. Here the landscape seems more erratic, and the regularity as well as the number of modes is less straightforward to determine. Yet,

---

[1]https://github.com/vdutor/SphericalHarmonics

---

**Algorithm 1** A cross-entropy-fitted importance sampling estimator.

---

1: Input: Measures $\mu$ and $\nu$, Number $N$ of points to be sampled, Budget fraction $r \in (0,1)$ to allocate to estimating the proposal.

2: Sample $X_1, \ldots, X_{\lfloor rn \rfloor}$ i.i.d. from the uniform measure on the sphere. Evaluate the integrand $f^{(p)}_{\mu,\nu}$ in 6 on them. Define
$$i_{\max} = \operatorname{argmax}\{f^{(p)}_{\mu,\nu}(X_i)|i \le \lfloor rn \rfloor\}.$$

3: Define $\hat{f}^{(p)}_{\mu,\nu}(x) = f^{(p)}_{\mu,\nu}(x)\mathbb{1}[\langle X_{i_{\max}}, x\rangle > 0]$, and evaluate the quantities
$$\varepsilon_{\lfloor rN \rfloor} = \frac{\sum_{i=1}^{\lfloor rN \rfloor} \hat{f}^{(p)}_{\mu,\nu}(X_i)X_i}{\|\sum_{i=1}^{\lfloor rN \rfloor} \hat{f}^{(p)}_{\mu,\nu}(X_i)X_i\|}, \quad R_{\lfloor rN \rfloor} = \frac{\|\sum_{i=1}^{\lfloor rN \rfloor} \hat{f}^{(p)}_{\mu,\nu}(X_i)X_i\|}{\sum_{i=1}^{\lfloor rN \rfloor} \hat{f}^{(p)}_{\mu,\nu}(X_i)}.$$

4: Let $\kappa_{\lfloor rN \rfloor} = \dfrac{R_{\lfloor rN \rfloor}(d - R^2_{\lfloor rN \rfloor})}{1 - R^2_{\lfloor rN \rfloor}}$ as in Sra (2012).

5: Sample $X_{\lfloor rN \rfloor + 1}, \ldots, X_N$ from $\frac{1}{2}(\operatorname{vmf}(\varepsilon_{\lfloor rN \rfloor}, \kappa_{\lfloor rN \rfloor}) + \operatorname{vmf}(-\varepsilon_{\lfloor rN \rfloor}, \kappa_{\lfloor rN \rfloor}))$.

6: Return
$$\frac{r}{\lfloor rN \rfloor}\sum_{i=1}^{\lfloor rN \rfloor} f^{(p)}_{\mu,\nu}(X_i) + 2\frac{1-r}{\lceil (1-r)N \rceil}\sum_{i=\lfloor rN \rfloor + 1}^{N} \frac{f^{(p)}_{\mu,\nu}(X_i)}{\operatorname{vmf}(X_i|\varepsilon_{\lfloor rN \rfloor}, \kappa_{\lfloor rN \rfloor}) + \operatorname{vmf}(X_i| -\varepsilon_{\lfloor rN \rfloor}, \kappa_{\lfloor rN \rfloor})}.$$

---

**Algorithm 2** The HKPV algorithm for sampling the harmonic ensemble of size $N$ Hough et al. (2006).

---

1: Input: $N$, the kernel $K$.

2: Sample $\theta_1 \sim \mathrm{d}\theta_1$ on the sphere

3: **for** $2 \le k \le N$ **do**

4:     Use rejection sampling to draw
$$\theta_k \sim \frac{K(\theta_k, \theta_k) - K_{k-1}(\theta_k, \theta_{1:k-1})\mathbf{K}^{-1}_{k-1}K_{k-1}(\theta_{1:k-1}, \theta_k)}{N - k + 1}\mathrm{d}\theta_k,$$

5:     where $K(\theta_k, \theta_{1:k-1}) = K(\theta_k, \theta_{1:k-1})^T = (K(\theta_k, \theta_1), \ldots, K(\theta_k, \theta_{k-1}))^T$, and

6:     $\mathbf{K}_{k-1} = (K(\theta_i, \theta_j))_{1 \le i,j \le k-1}$

7: **end for**

8: Return $\{\theta_1, \ldots, \theta_N\}$

---

**Algorithm 3** The SHCV method Leluc et al. (2024)

---

1: Input: $N$, spherical harmonics $(\varphi_l)_{l=1}^s$.

2: Sample $\theta_1, \ldots, \theta_N$ i.i.d. uniformly on the sphere.

3: Compute the matrices $\boldsymbol{\Phi} = (\varphi_l(\theta_i))_{1 \le i \le N, 1 \le l \le s}$, $\mathbf{H} = \boldsymbol{\Phi}^T\boldsymbol{\Phi}$

4: Compute the vector $\mathbf{f}_N = (f_{\mu,\nu}(\theta_i))_{1 \le i \le N}$

5: Let $\Pi = \mathbf{H} \cdot (\mathbf{H}^T\mathbf{H})^{-1} \cdot \mathbf{H}^T$ and $\mathbf{1}_N = (1, \ldots, 1)^T$

6: Return $SHCV^p_N(\mu, \nu) = \frac{\mathbf{1}^T_N(I_N - \Pi)\mathbf{f}_N}{\mathbf{1}^T_N(I_N - \Pi)\mathbf{1}_N}$

---

intuitively the resulting integrands should have a relatively sparse decomposition in the bases of spherical harmonics, as confirmed by Figure 6.

---

**Algorithm 4** Crude Monte Carlo integration with a single control variate Nguyen & Ho (2024)

1: Input: $N$, a single control variate $\varphi : \mathbb{S}^{d-1} \to \mathbb{R}$
2: Sample $\theta_1, \dots, \theta_N$ i.i.d. uniformly on the sphere.
3: Compute $f_N = (f_{\mu,\nu}(\theta_i))_{1 \le i \le N}$, $\mu_N = \frac{1}{N} \sum\limits_{i=1}^{N} f_{\mu,\nu}(\theta_i)$
4: Compute $\beta = \dfrac{\sum\limits_{i=1}^{N} (f_{\mu,\nu}(\theta_i) - \mu_N)\varphi(\theta_i)}{\sum\limits_{i=1}^{N} (f_{\mu,\nu}(\theta_i) - \mu_N)^2}$
5: Return $\frac{1}{N} \sum\limits_{i=1}^{N} (f(\theta_i) - \beta\varphi(\theta_i))$

---

**Algorithm 5** Sampling from the Haar measure (Meckes, 2019, Paragraph 1.2)

1: Sample $X_1, \dots, X_d$ i.i.d. $\mathcal{N}(0, I_d)$.
2: Compute the QR decomposition of $\mathbf{X} = (X_1 | \dots | X_d)$ via the Gram-Schmidt algorithm.
3: Return the matrix $Q$.

---

### A.5 A few words on repelled point processes

The repelled point processes behave inconsistently in the experiments of Section 6. Intuitively, a small repulsive perturbation, meaning a small $\epsilon > 0$ in (19), should lead to some variance reduction, yet the magnitude of $\epsilon$ seems to depend on the number $N$ of points to repel as well as on their distribution. In this section, we experimentally investigate this optimal choice for $\epsilon$, to guide future theoretical investigations. Following the seminal case of a homogeneous Poisson process in $\mathbb{R}^d$, we expect variance reduction to happen when $\epsilon$ is of the order of $1/N$, where $N$ is the cardinality of the configuration to repel.

To assess the influence of $\epsilon$, we sampled 100 independent realizations of the repelled estimator for each of a discrete set of values of $\epsilon$ and a choice of integrands and initial point processes, and reported the $\chi^2$ confidence interval for the variance of each estimator. We correct these confidence intervals with a Bonferroni correction across the finite set of values for $\epsilon$, to reach a simultaneous confidence level of $0.969$.

Figure 10 shows the results for an initial point process made of i.i.d. uniform points on the sphere, for a choice of indicators and values of $N$. The red line is a reference corresponding to $\epsilon = 0$, placed at an arbitrary low value of $\epsilon$ for comparison. The gray dashed line corresponds to $\epsilon = 1/N$. We observe that $\epsilon = 1/N$ is indeed a sensible choice, leading to a variance reduction by a factor up to $2$, even for a non-smooth integrand like the indicator of a half-sphere.

Using the i.i.d. repelled points to build the SHCV estimator, as denoted by *Repelled SHCV* in Section 6, the situation becomes much more unstable, as shown in Figure 11. Looking closely, it seems that a choice of $\epsilon$ slightly under $1/N$ seems to consistently lead to some variance reduction, but a small variation in $\epsilon$ can have drastic consequences, as observed on Figure 11b. Note also that, unlike the vanilla repelled i.i.d. estimator, the optimal choice for $\epsilon$ seems to be dimension-dependent, as shown by the difference in the dips between Figure 10b and Figure 11e. Hence, although repelling the points in the use of *SHCV* has the potential to diminish the variance, further theoretical investigation are required to correctly tune $\epsilon$.

Finally, in line with the experimental observations of Hawat et al. (2023), repelling structured points can also lead to a straight-up increase in the variance. This is the case when the starting configuration is a randomized grid, as in the *QMC* or the *UnifOrtho* estimators in Section 6, see Figure 12. Yet, Figure 12b suggests that *Repelled UnifOrtho* can actually lead to a dip in variance when integrating an indicator. This was an unexpected behavior and further theoretical work is needed to understand this phenomenon.

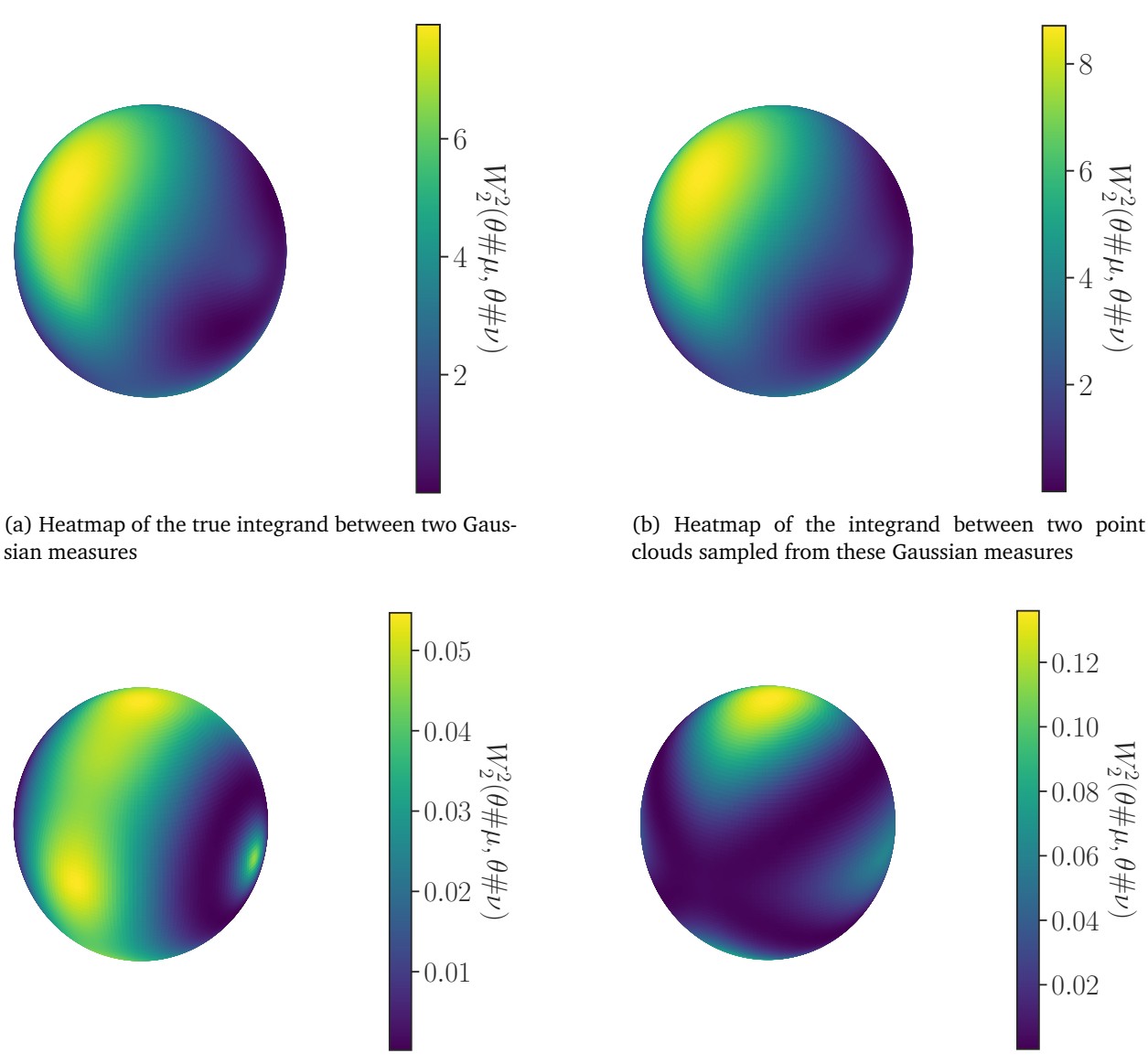

(a) Heatmap of the true integrand between two Gaussian measures

(b) Heatmap of the integrand between two point clouds sampled from these Gaussian measures

(c) Heatmap of the integrand between point clouds #2 and #34

(d) Heatmap of the integrand between point clouds #3 and #35

Figure 9: Heatmaps of the integrand in 3D for various distributions.

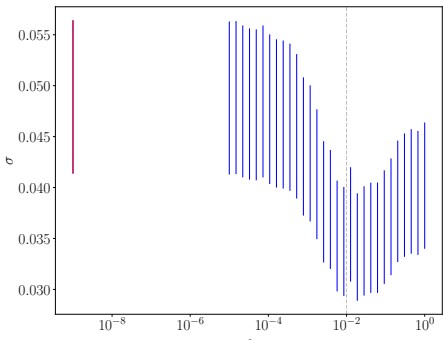

(a) Integration of the indicator of a half-sphere with $N = 100$ i.i.d. points, $d = 3$

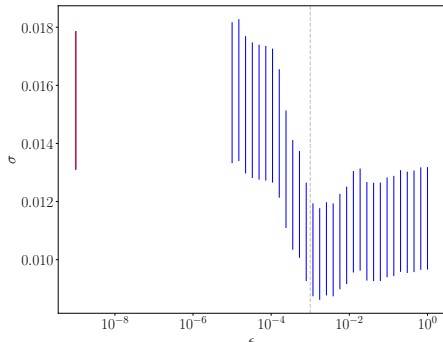

(b) Integration of the indicator of a half-sphere with $N = 1000$ i.i.d. points, $d = 3$

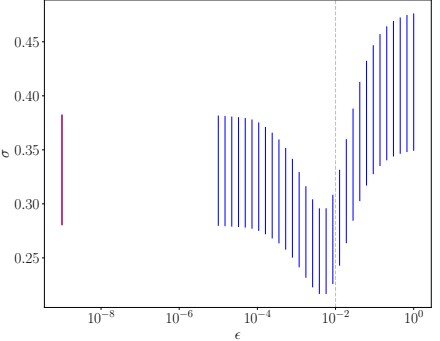

(c) Integration of the sliced Wassertein between sampled Gaussian supported on 100 points with $N = 100$ i.i.d. points, $d = 3$

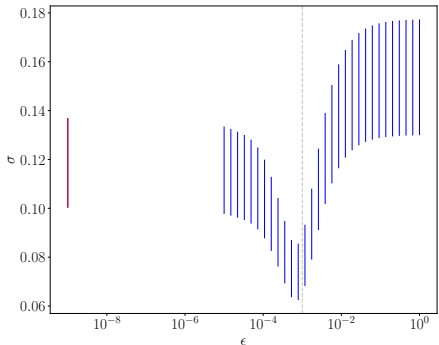

(d) Integration of the sliced Wassertein between sampled Gaussian supported on 100 points with $N = 1000$ i.i.d. points, $d = 3$

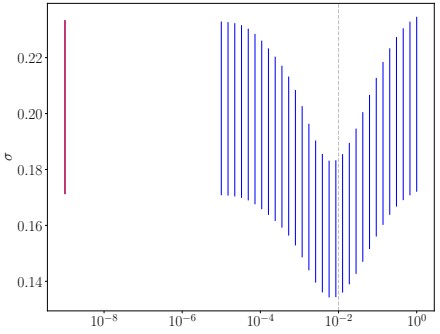

(e) Integration of the sliced Wassertein between sampled Gaussian supported on 100 points with $N = 100$ i.i.d. points, $d = 10$

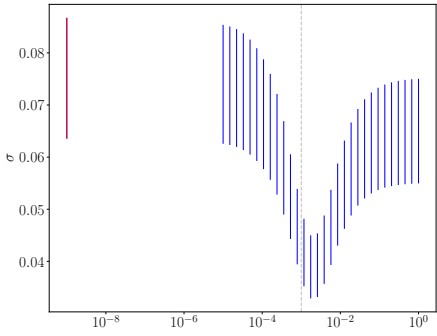

(f) Integration of the sliced Wassertein between sampled Gaussian supported on 100 points with $N = 1000$ i.i.d. points, $d = 10$

Figure 10: Confidence intervals for the variance of the estimator with a Bonferroni corrected confidence level of 0.969, where the repelled points are sampled i.i.d.

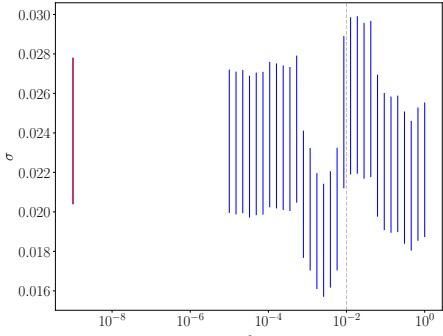

(a) Integration of the indicator of a half-sphere with $N = 100$ and SHCV, $d = 3$

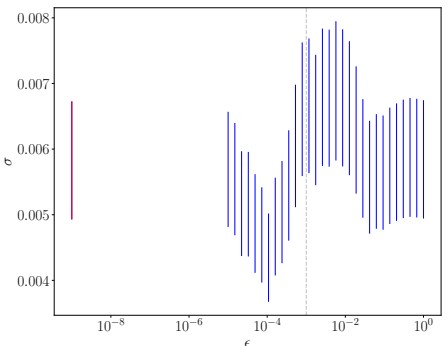

(b) Integration of the indicator of a half-sphere with $N = 1000$ and SHCV, $d = 3$

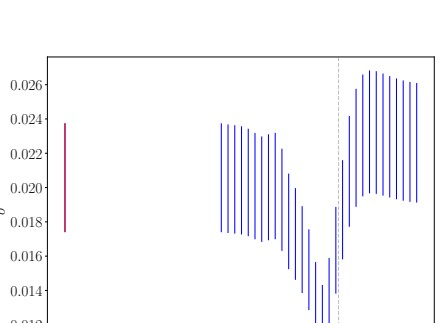

(c) Integration of the sliced Wassertein between sampled Gaussians supported on 100 points with $N = 100$ and SHCV, $d = 3$

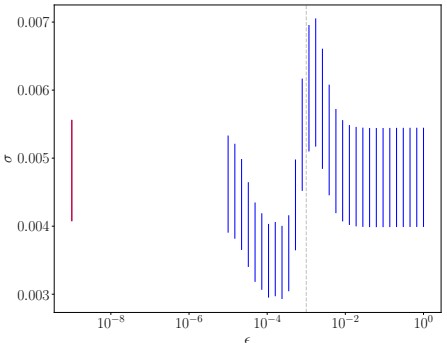

(d) Integration of the sliced Wassertein between sampled Gaussians supported on 100 points with $N = 1000$ and SHCV, $d = 3$

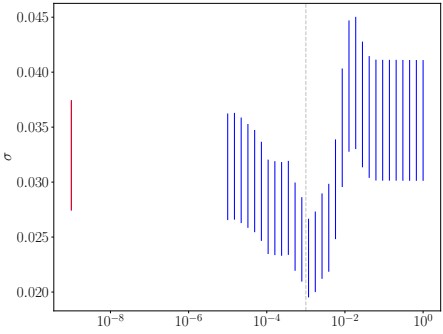

(e) Integration of the sliced Wassertein between sampled Gaussians supported on 100 points with $N = 1000$ and SHCV, $d = 10$

Figure 11: Confidence intervals for the variance of the estimator with a Bonferroni corrected confidence level of 0.969 where the SHCV estimator is built on repelled i.i.d. uniform points.

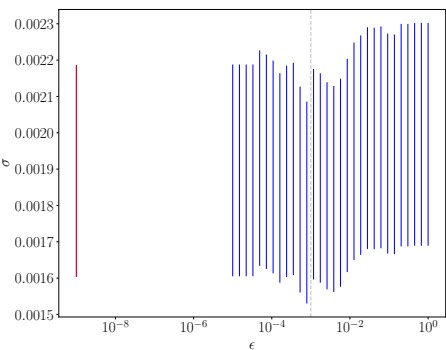

(a) Integration of the indicator of a half-sphere with $N = 1000$ RQMC points, $d = 3$

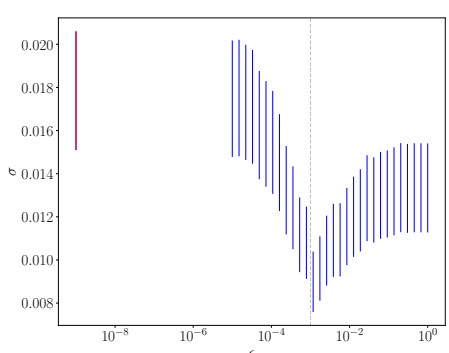

(b) Integration of the indicator of a half-sphere with $N = 1000$ UnifOrtho points, $d = 3$

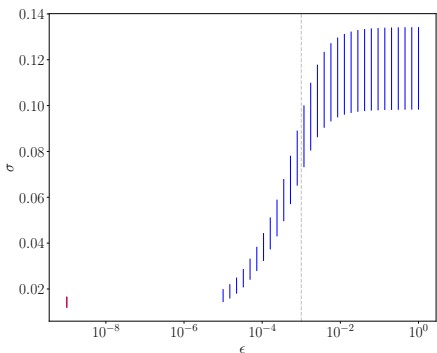

(c) Integration of the sliced Wassertein between sampled Gaussian supported on 100 points with $N = 1000$ UnifOrtho points, $d = 3$

Figure 12: Confidence intervals for the variance of the estimator with a Bonferroni corrected confidence level of 0.969, when repelling points sampled using either UnifOrtho or RQMC in $d = 3$.

