# OpenReview forum: "Repulsive Monte Carlo on the sphere for the sliced Wasserstein distance"
_TMLR — Accepted by TMLR_

### Review · Reviewer_AZwZ · 2025-10-08

**Summary Of Contributions:**

This paper investigates the use of repulsive Monte Carlo methods for the numerical integration required to compute the sliced Wasserstein distance. The core contribution is a unified comparison of various quadrature techniques, focusing on those that enforce "repulsion" between sampling points on a high-dimensional sphere. The authors introduce and benchmark methods based on DPPs and repelled point processes. Another contribution is a novel theoretical analysis of the variance of the UnifOrtho estimator, which leverages spherical harmonics to explain its strong empirical performance in high dimensions.

---

**Strengths:**

- The paper is well-written and provides a clear guide for approximating the integrand.
- The empirical study is thorough, comparing a wide and relevant range of methods across multiple dimensions and tasks.
- The new variance analysis for the UnifOrtho estimator is a strength. It provides a deeper theoretical understanding of the variation reduction feature of the UnifOrtho algorithm.

---

**Weaknesses:**

- Although the paper is framed around repulsive Monte Carlo, its survey and experimental comparisons are not exhaustive. The authors focus primarily on DPPs and a specific potential-based repulsion scheme. Other relevant algorithms from this domain are not discussed or benchmarked, such as methods based on Wang-Landau, Stein's method, or adaptive biasing potential method.
- While the investigation is valuable, the newly benchmarked "Repelled" processes are shown to offer nearly negligible performance difference among them, e,g., in Figure 7 and Figure 8.
- The DPP-based methods are demonstrated to be computationally prohibitive in higher dimensions, limiting their practical applicability for this task. Thus, the scope of this paper sounds a bit narrowed, ignoring the comparisons to other repulsive Monte Carlo methods.

**Audience:**

Yes

**Audience Explanation:**

The findings of this paper would be of significant interest to a substantial portion of the TMLR audience, particularly those who work with generative models, optimal transport, and high-dimensional probability distributions.

The paper directly addresses a practical, foundational problem: how to efficiently and accurately estimate the sliced Wasserstein, which is a popular loss function for training GANs, VAEs, and one-dimensional OT.

**Claims And Evidence:**

Yes

**Claims Explanation:**

Proposition 4 provides a precise, closed-form expression for the variance of the UnifOrtho estimator. This formula directly links the variance to the spherical harmonic coefficients of the function being integrated.

The central, practical claim, i.e., practitioners should use randomized QMC in low dimensions and UnifOrtho in high dimensions, is backed by a comprehensive set of experiments in Section 6.

**Requested Changes:**

1. The paper frames its contributions around "repulsive Monte Carlo" but primarily focuses on DPPs and a specific potential-based repulsion scheme. The work would be strengthened by briefly acknowledging the broader literature on repulsive and adaptive sampling in the introduction or related work section (Section 2). Mentioning other prominent approaches, such as [1,2,3,4], and briefly justifying the paper's focus on DPPs would make the survey feel more complete and avoid questions about why certain methods were excluded from the benchmark. This does not require new experiments, but rather a few sentences to better posit the work.

> [1]. Mao Ye, Tongzheng Ren, and Qiang Liu. "Stein self-repulsive dynamics: Benefits from past samples." Advances in Neural Information Processing Systems 33 (2020): 241-252.
>
> [2]. Gersende Fort, et al. "Convergence of the Wang-Landau algorithm." Mathematics of Computation 84.295 (2015): 2297-2327.
>
> [3]. Michel Benaïm, and Charles-Édouard Bréhier. "Convergence of adaptive biasing potential methods for diffusions." Comptes Rendus. Mathématique 354.8 (2016): 842-846.
>
> [4]. Gersende Fort, et al. "Convergence and efficiency of adaptive importance sampling techniques with partial biasing." Journal of Statistical Physics 171.2 (2018): 220-268.

2. The finding that the simple repulsive method provides only moderate variance reduction is an interesting result. The paper would benefit from a short discussion in the conclusion speculating on why this might be the case. Is it a fundamental limitation of this specific energy function, or could more sophisticated repulsion schemes potentially yield better results? This would add valuable insight for future work in this direction.

---

### Review · Reviewer_TXUM · 2025-10-25

**Summary Of Contributions:**

This paper proposes new empirical methods of estimating the Sliced Wasserstein distance and does a detailed comparative study of it.

**Additional Comments:**

N/A

**Audience:**

Yes

**Audience Explanation:**

Although this paper is not my area specialization, from the introduction given I am fairly confident that this topic can be of interest to many people - if the paper is extensively revised to be a much better writing.

**Claims And Evidence:**

No

**Claims Explanation:**

**strengths**

The experimental study is fairly detailed.

**weakness**

The paper is quite badly written - particularly so, considering that its a submission to a ML journal and hence the baseline expectation would be to expect a clear connection to ML being made.

Apart from the general issue above, the specification of the different algorithmic techniques being presented is quite vague. Its hard to distinguish between which parts are general commentaries and which parts are specification of the algorithm to be tested.

**Requested Changes:**

All my requests below are essential to be addressed for securing acceptance.

- The second paragraph of page-3 gives some details on how computing the sliced-Wasserstein (SW) distance impacts other conventional aspects of ML. But this is too thin a description and motivation for readers like me, who are ML theory/optimization theory specialists, to understand the need and scope of this question. My first and foremost suggestion is that this needs to be expanded into a detailed section explaining the interface between usual ML questions and SW distance. The beginning of the paper needs to be a lot more ML-friendly.

  What things in ML can improve if we have better estimators of SW distance?

- The section 4 on new candidate estimators is quite unclear. For a start we need an explicit definition of the von-Mises-Fisher distribution. Can this be organized into separate pseudocodes for computing SW via each of the proposed new methods of sampling points on the sphere?

- The situation with the UnifOrtho seems a bit unclear. The discussion in section 3.2.2 seems to suggest that UnifOrtho is not a good estimator. Then given all these discussions why do we have a separate Section 5 dedicated to discussing it? Isnt that discussion in Section 5 better suited as an appendix?

- The beginning of Section 6 needs a lot of reorganizing. I think this bullet point list is very hard to parse, particularly because multiple methods are being referred to the same previous subsection. What we need are separate numbered equations and pseudocode for each of the methods being tested and those be referred to from this list to indicate the methods being tested.

-  Lastly, for each of the experiments, we need to see a final comparison table of performance between the different techniques which makes it clear as to who won that race and on what metrics. And there needs to be a final reflection on what methods are being recommended by the authors for the readers. It is to be expected that the final recommendation would depend on the ML question of the end-user, but that dependency is hardly becoming clear.

---

> ### Author Response · Authors · 2025-11-20
> **Response to Reviewer TXUM**
>
> Thank you for your clear feedback and suggestions.
>
> About your requested changes:
> 1. We will dedicate a separate subsection of the introduction to expanding on the benefits for ML of an accurate estimation of the sliced Wasserstein distance. Besides developing the examples we already mention, like Bonet et al.'s generative models that are based on a sliced Wasserstein gradient flow. We propose to add a few lines on other ML applications to autoencoders [Kolouri et al. 2018](https://arxiv.org/abs/1804.01947), texture synthesis [Heitz et al 2020](https://arxiv.org/abs/2006.07229), differential privacy [Rakotomamonjy et al. 2021](https://arxiv.org/abs/2107.01848), and more recently to gradient flows on manifolds [Bonet et al. 2024](https://arxiv.org/abs/2403.06560). In all these cases, the sliced Wasserstein is used as a way to quantify the distance between two probability distributions $\mu$ and $\nu$, which can only be accessed through samples. There are two natural sources of error in estimating the sliced Wasserstein distance between $\mu$ and $\nu$, and both errors will propagate in the overarching ML pipeline. One source of error is related to replacing $\mu$ and $\nu$ by their empirical counterparts $\hat{\mu}$ and $\hat{\nu}$ [Manole et al. 2019](https://arxiv.org/abs/1909.07862), and the other is the error in estimating the sliced Wasserstein distance between $\hat{\mu}$ and $\hat{\nu}$. Our paper focuses on reducing this second source of error, which can be done rather independently of the initial ML problem.
>
> 2. The von-Mises Fisher is succintly defined in the definition of
>  $G_{\mathrm{vmf}}$ in Section 4.1.
> We propose to extract the definition of the von-Mises-Fisher distribution and state it before the definition of $\mathcal{G}_{\mathrm{vmf}}$. We will also add an appendix with pseudocode for the main sampling strategies.
>
> 3. Upon re-reading, our Section 3.2.2 indeed seems to be negative about UnifOrtho, but we only mean at this stage of the paper that there are no theoretical result supporting any variance reduction. There even are counterexamples of empirical measures where, if one estimates the sliced Wasserstein with UnifOrtho, one obtains a variance that is *provably larger* than the variance of the naive i.i.d. estimator of the sliced Wasserstein distance. This indeed would intuitively discard UnifOrtho at first glance. Yet, in our numerical benchmark, as well as in the experimental sections of [Sissouk et al. 2025](https://arxiv.org/abs/2502.02275), [Rowland et al. 2019](https://arxiv.org/abs/1903.03784), UnifOrtho shines as an effective sampling strategy in dimension $d\geq20$. This is why we dedicated some effort to understanding the variance of UnifOrtho-based estimators in Section 5, providing a formula that explains both the counterexamples and the numerical success of UnifOrtho in estimating the sliced Wasserstein distance in large dimensions. We believe this analysis is valuable, as also noted by Reviewer XjpC and Reviewer AZwZ.
>
> 4. We will reorganize our different paragraphs in Section 6. For instance, each bullet point of the initial list will refer to a unique equation or paragraph inside our paper.
>
> 5. On this matter, we think that graphs as presented are the clearest way to describe our results. Indeed, multiple parameters are varying and graphs give a strong visual interpretation, while tables would be too large and hard to read. As mentioned in our first point, it is not that surprising that the method of choice only loosely depends on the ML task performed, as we are focussing here on estimating the sliced Wasserstein distance between two empirical distributions. We will strive to make this clearer in our conclusion.

---

> ### Comment · Reviewer_TXUM · 2026-01-07
> **Thanks!**
>
> Thanks for the extensive edits in the draft.

---

### Review · Reviewer_XjpC · 2025-11-09

**Summary Of Contributions:**

The authors evaluate the Monte-Carlo method for computing the sliced Wasserstein distance (SWD), which can be used as a proxy to Wasserstein distance that is widely used in OT but always intractable especially in the high-dimensional setting. The contributions include benchmark randomized quadratures for estimating SWD; after identifying the empirically most successful MC based estimator, the authors further provide theoretical understanding of of the concentration in terms of the dimension $d$ and the basis $Y^\ell_k$.

**Audience:**

Yes

**Audience Explanation:**

The introduction part of the paper is very informative to the readers in the related area. The simulation part is also comprehensive in comparing the existing MC based approaches for computing SWD, which can be used as a guideline to pick the good strategy.

**Claims And Evidence:**

Yes

**Claims Explanation:**

I have the following concerns and questions;

1. Can you explain more on the notations in Proposition and make it self-contained? For instance, the definition of $l, h_l,k$ in describing the basis.
2. My main concern is on Proposition 4, to me it is essentially a bias-variance tradeoff, where the proof starts with the variance-covariance-bias expression of $Var(\frac{1}{d}\sum^d_{i=1} f(X_i))$, in the proof, there are some steps not clear to me:

i) How the author obtain the decomposition of $f(x) = \sum_{\ell = 0}^{\infty} \sum_{k = 1}^{h_\ell} \widehat{f}(\ell,k)\, Y_k^{\ell}(x)$, any assumption is made during the decomposition?

ii) by integrating the decomposition of $f$ to (21), can the authors provide more details on the simplification? Because to me, it is not that obvious to get $\mathbb{E}\big[f(X_{1}) f(X_{2})\big] = \sum_{\ell = 0}^{\infty} (-1)^{\ell} \lambda_{2\ell} \, \mu_{2\ell}(f)$ directly.

3. The comment on the Proposition 4 is intuitive and can also be supported by some simulation results. When the authors justify that 'When the dimension $d-2$ grows, the generalized Stirling formula yields  $\lambda_{2\ell} = \mathcal{O}(\ell^{-\frac{d-2}{2}})$, so that only the first terms in either (19) or (20) carry significant weight.', if I understand correctly, when dimension increase, the variance of UnifOrtho estimator will be close that that of crude MC, can this also be supported in the simulation?

**Requested Changes:**

Each section and paragraph especially when previous approach in the literature is introduced, the notation as well as its explanation is supposed to be self-contained, the current version is a bit hard to follow since the notations are repeatedly used across sections, and many important results are being referred to the original paper. It would make the paper more readable if the authors can summarize and compare the existing MC based approach in terms of the variance reduction rate to give a more straightforward comparison and high-level understanding of these approaches.

---

> ### Author Response · Authors · 2025-11-20
> **Response to Reviewer XjpC**
>
> Thanks for your thorough review.
>
> 1. Concerning your technical questions:
>
> 1.1. We will make the proposition self-contained.
>
> 1.2.i. This decomposition is the $L^2$ decomposition of $f$ in the basis of spherical harmonics. The spherical harmonics are specifically designed to be a Hilbert basis of the space $L^2$ of square-integrable functions on the sphere. The decomposition thus holds in the $L^2$ sense. We give a very quick reminder on this basis in Appendix A, but also refer to ([Dai and Xu 2013](https://arxiv.org/abs/1304.2585) for a detailed discussion, as well as [Leluc et al 2024](https://arxiv.org/pdf/2402.01493) Proposition 1).
>
> 1.2.ii. We propose to add the following details to further explain.
> The Funk operator is linear and continuous over $L^2$. Using the decomposition of $f$ on the spherical harmonics basis, it holds that, in the $L^2$ sense,
>
> $$
>     \mathcal{F}(f) = \sum_{l = 0}^\infty (-1)^l \lambda_{2l} \sum_{k = 1}^{h_l} \hat{f}(k,l) \mathrm{Y}_k^l.
> $$
>
> Plugging into (21) and applying the Parseval identity, and since $\mathbb{E}[f(X_1)f(X_2)] = (f,\mathcal{F}f)_{L^2}$ , we obtain
>
> $$
>     \mathbb{E}[f(X_1)f(X_2)] = \sum_{l = 0}^\infty (-1)^l \lambda_{2l} \sum_{k = 1}^{h_l} (\hat{f}(k,l))^2,
> $$
>
> concluding the derivation.
>
> 1.2.iii. Actually, when the dimension grows, the variance is close to
>
> $$
>     \frac{1}{d}\operatorname{Var}[f(X_1)] - \frac{1}{d} \mu_2(f),
> $$
>
> since only the first term of the sum in (19) carries significant weight. This term cannot be ignored since it is of the same magnitude ($1/d$) as the i.i.d. variance term.
>
> 2. As stated in our answer to reviewer TXUM, we will make sure that each bullet refers to a single method; this will indeed clarify Section 6. Following your recommendation, we will also make sure that each section is self-contained in terms of notation.
>
> 3. We will also add a small recap ahead of the experimental section, summarizing the theoretical guarantees offered by the different methods presented.

---

### Decision · Action_Editor_jU7S · 2026-01-27

**Recommendation:** Accept as is

**Additional Comments:**

Please remember to remove the table of contents. Also, please capitalize the first letter of each word in the title.

**Audience:**

Yes

**Audience Explanation:**

MCMC-based sampling methods are of broad interest in generative AI and machine learning, and the audience of TMLR would be interested in the paper’s findings.

**Claims And Evidence:**

Yes

**Claims Explanation:**

This paper studies Monte Carlo methods for integrating functions over the unit sphere, with a focus on computing the sliced Wasserstein distance, an integral over random projections that is widely used in machine learning. The claims in the submission are supported by a combination of theoretical analysis and empirical experiments.